# Insights into evapotranspiration partitioning based on hydrological observations using the generalized proportionality hypothesis

Amin Hassan[1], I. Colin Prentice[2,3], Xu Liang[1]

[1]Department of Civil and Environmental Engineering, University of Pittsburgh, Pittsburgh, Pennsylvania, 15213, USA
[2]Georgina Mace Centre for the Living Planet, Department of Life Sciences, Imperial College London, Silwood Park Campus, Ascot, UK
[3]Department of Earth System Science, Ministry of Education Key Laboratory for Earth System Modeling, Institute for Global Change Studies, Tsinghua University, Beijing, China

*Correspondence to*: Xu Liang (xuliang@pitt.edu)

**Abstract.** Evapotranspiration comprises transpiration, soil evaporation, and interception. The partitioning of evapotranspiration is challenging due to the lack of direct measurements and uncertainty of existing evapotranspiration partitioning methods. We propose a novel method to estimate long-term mean transpiration to evapotranspiration ($E_t/E$) ratios based on the generalized proportionality hypothesis using long-term mean hydrological observations at the watershed scale. We tested the method using 648 watersheds in the United States classified into six vegetation types. We mitigated impacts of the variability associated with different $E_p$ data products by rescaling their original $E_p$ values using the product $E/E_p$ ratios in combination with the observed $E$ calculated from watershed water balance. With $E_p$ thus rescaled, our method produced consistent $E_t/E$ across six widely used $E_p$ products. Shrubs (0.38) and grasslands (0.33) showed lower mean $E_t/E$ than croplands (0.46) and forests (respectively 0.73, 0.55, and 0.68 for evergreen needleleaf, deciduous broadleaf, and mixed forests). $E_t/E$ showed significant dependence on aridity, leaf area index, and other hydrological and environmental conditions. Using $E_t/E$ estimates, we calculated transpiration to precipitation ratios ($E_t/P$) ratios and revealed a bell-shaped curve at the watershed scale, which conformed to the bell-shaped relationship with the aridity index (AI) observed at the field and remote-sensing scales (Good et al., 2017). This relationship peaked at an $E_t/P$ between 0.5 and 0.6, corresponding to an AI between 2 and 3 depending on the $E_p$ dataset used. These results strengthen our understanding of the interactions between plants and water and provide a new perspective on a long-standing challenge for hydrology and ecosystem science.

## 1 Introduction

Partitioning evapotranspiration is important for understanding water and energy balances of terrestrial ecosystems. Evapotranspiration has been predicted to increase at the expense of soil moisture due to climate change (Li et al., 2022; Niu et al., 2019) with potential implications for future projections of water, energy, and carbon balances. Large uncertainty remains in the partitioning of evapotranspiration into its components: transpiration, interception, and bare soil evaporation. Various methods have been developed to partition evapotranspiration based on measurements (Kool et al., 2014; Stoy et al., 2019).

These include (1) flux-variance similarity methods using high frequency (10–20 Hz) flux tower measurements, which estimate $E_t/E$ based on carbon-water correlation since transpiration and plant carbon uptake are concurrent (Scanlon and Kustas, 2010, 2012; Scanlon and Sahu, 2008; Skaggs et al., 2018); (2) eddy-covariance methods, which estimate $E_t/E$ using assumptions related to water use efficiency based on widely available half-hourly/hourly eddy covariance measurements (Berkelhammer et al., 2016; Li et al., 2019; Scott and Biederman, 2017; Yu et al., 2022; Zhou et al., 2016); and (3) isotopic methods (Griffis, 2013; Williams et al., 2004; Zhang et al., 2011). Measurements of sap flow through plant stems have also been commonly used to more directly estimate transpiration. Sap flow measurements are classified into three groups (Kool et al., 2014): heat balance methods (Čermák et al., 1973; Sakuratani, 1981, 1987), heat pulse methods (Cohen et al., 1981; Green et al., 2003; Swanson and Whitfield, 1981), and constant heater methods (Čermák et al., 2004; Granier, 1985). Poyatos et al. (2021) compiled 202 sap flow datasets to form the global SAPFLUXNET dataset. Recent studies have used remotely sensed solar-induced fluorescence (SIF) measurements (Alemohammad et al., 2017; Damm et al., 2018; Liu et al., 2022; Lu et al., 2018; Pagán et al., 2019; Shan et al., 2019) to estimate global transpiration, relying on the close coupling between transpiration and photosynthesis.

The ratio of transpiration to evapotranspiration ($E_t/E$) is a particularly important quantity because the controls on T (which is tightly regulated by plants through stomatal behavior) are substantially different from the controls on the other two components. The evapotranspiration partitioning methods summarized above have multiple limitations and produce an alarmingly wide range of values for the global mean $E_t/E$. Wei et al. (2017) showed mean global $E_t/E$ varying from 0.24 to 0.90 based on a variety of remote-sensing, isotopic, and modelling studies. Another compilation by Liu et al. (2022) showed the mean varying between 0.24 and 0.86. Schlesinger and Jasechko (2014) showed that $E_t/E$ ratios derived from isotopic methods tend to be systematically higher than those produced by other methods. It has also been shown that two different evapotranspiration partitioning methods could produce greatly different $E_t/E$ values at the same site (Cavanaugh et al., 2011; Moran et al., 2009). Some $E_t/E$ estimates at the stand scale ignore transpiration from subcanopy vegetation, resulting in underestimation (Schlesinger and Jasechko, 2014). There is no consensus on which method is more accurate (Stoy et al., 2019); this presents a challenge for applying the $E_t/E$ estimates using any of the above methods, especially when they are developed based on data at site scale but are applied at larger (regional to global) spatial scales.

Few studies have considered partitioning evapotranspiration based on hydrological concepts using widely available long-term hydrological observations, which could in principle provide reliable methods to estimate $E_t/E$. Gerrits et al. (2009) estimated monthly and (upscaled) annual transpiration based on precipitation, interception, soil moisture, and the aridity index. They estimated $E_t/E$ by modeling interception (which includes topsoil evaporation) as a daily threshold process (threshold is the interception storage capacity) and used rainfall distributions to upscale it to the monthly and then annual interception. Transpiration was modeled as a monthly threshold process based on net rainfall (precipitation minus interception), with the threshold being the soil moisture storage estimated based on a hydrological model, and upscaled it to annual transpiration via a rainfall distribution. $E_t/E$ is then calculated by assuming evapotranspiration is interception plus transpiration, since topsoil evaporation is included in interception, and deeper soil and open water evaporations are neglected. Mianabadi et al. (2019)

extended their approach and applied it globally. In this study, we propose a new method to partition evapotranspiration based on the Generalized Proportionality Hypothesis (GPH) using long-term hydrological observations. The GPH was initially used by the United States Soil Conservation Service (SCS) for runoff calculation (USDA SCS, 1985) and was afterwards generalized by Ponce and Shetty (1995a, b). Wang and Tang (2014) provided a comprehensive discussion of the use of GPH and noted its connection to various models, including the "abcd" model, the SCS direct runoff model, and the Budyko-type models. The GPH partitions water fluxes into their components and has been implemented as a two-stage partitioning. The first stage partitions precipitation into soil wetting and surface runoff; the second stage partitions soil wetting into baseflow and evaporation (Ponce and Shetty, 1995a, b; Tang and Wang, 2017). We follow an approach based on the GPH partitioning of soil wetting to estimate catchment $E_t/E$ based on hydrological observations. Due to the wider availability of hydrological observations compared to the observations required for the techniques previously mentioned, this method has a wide potential for application in gauged watersheds across the globe.

The objectives of our study are: 1) to develop a new method to estimate $E_t/E$ at the catchment scale based on long-term hydrological observations, 2) to test the method and evaluate its robustness to different data products using watersheds with different vegetation types, 3) to find $E_t/P$ (transpiration/precipitation) ratios based on $E_t/E$ and to compare this to previous studies, and 4) to understand the effect of hydrological and environmental conditions on both $E_t/E$ and $E_t/P$. The paper is organized as follows. Section 2 describes the newly developed method and the datasets used. Section 3 investigates the differences in $E_p$ data products, and the use of a rescaled $E_p$ for $E_t/E$ estimation. Section 4 presents the results from the new method. Section 5 discusses the results and investigates their dependence on hydrological and environmental factors. Section 6 provides an insight into the variation of some existing partitioning methods. Section 7 summarizes our conclusions.

## 2 Methods and Data

### 2.1 Theory

We present a new method to estimate long-term mean $E_t/E$ ratios at a watershed scale by taking advantage of long-term available hydrological observations. The new method is based on the Generalized Proportionality Hypothesis (GPH), shown in equation (1). the GPH equation has been previously established in the literature based on the observed relationships found by L'vovich (1979) and the later mathematical derivation (and generalization) by Ponce and Shetty (1995a, b). The proportionality hypothesis of the SCS method was obtained based on observed data from a larger number of watersheds (USDA SCS, 1985), which was then generalized by Ponce and Shetty (1995a). GPH partitions an unbounded water quantity $Z$ into an unbounded water quantity $Y$ and a water quantity $X$ that is bound by its potential value $X_p$. The value $X_0$ is the initial quantity of $X$ that is fulfilled prior to the competition between $X$ and $Y$; for example, interception is a portion of E that is initially lost and not accessible for baseflow:

$$\frac{X - X_0}{X_p - X_0} = \frac{Y}{Z - X_0}$$ (1)

Ponce and Shetty (1995a, b) applied the GPH for hydrological partitioning. They partitioned annual precipitation over two stages: the first stage partitions precipitation into catchment wetting and surface runoff; and the second stage partitions wetting (W) into evapotranspiration (E) and baseflow (Qb) as shown in Figure 1. Both stages of partitioning follow the generalized formula in equation (1). The two-stage partitioning is well established, has been proved with thermodynamic principles (Wang et al., 2015), and has been extensively used in the literature in studies (Abeshu and Li, 2021; Chen and Wang, 2015; Sivapalan et al., 2011; Tang and Wang, 2017; Wang and Tang, 2014).

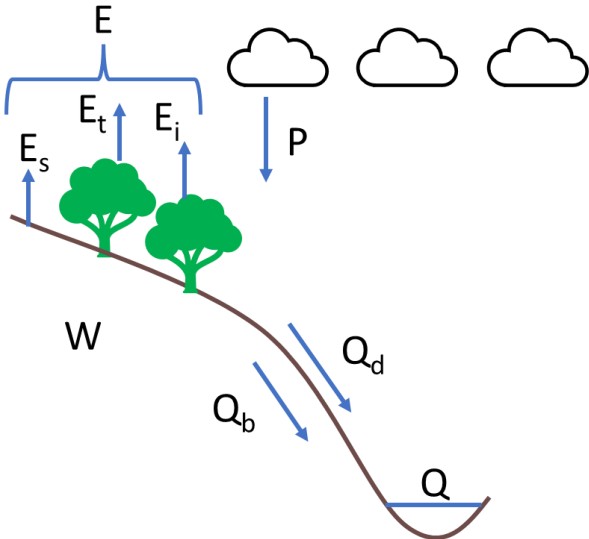

**Figure 1: Two stage partitioning of annual precipitation. E: evapotranspiration; Es: soil evaporation; Ei: interception evaporation; Et: transpiration; P: precipitation; W: soil wetting; Qb: baseflow; Qd: direct runoff; Q: total runoff.**

In this work, we use the second stage partitioning to partition wetting into evapotranspiration and baseflow as shown in equation (2):

$$\frac{E - E_0}{E_p - E_0} = \frac{Q_b}{W - E_0}$$ (2)

where $E_0$ is the initial evapotranspiration that does not compete with baseflow and $E_p$ is the potential evapotranspiration. $W$ can be estimated from watershed balance as $P - Q_d$, where $P$ is precipitation and $Q_d$ is direct runoff. $E$ can be estimated from watershed balance as $P - Q$, where $Q$ is the total runoff (since the long-term mean soil moisture change can be ignored). Initial evapotranspiration ($E_0$) has been represented in different ways in the literature. Ponce and Shetty (1995a, b) and Sivapalan et al. (2011) used $\lambda E_p$ to represent $E_0$, where $\lambda$ is a coefficient, Tang and Wang (2017) and Wang and Tang (2014) used $\lambda W$, and Abeshu and Li (2021) used $\lambda E$. In this study, we choose $\lambda E$ as $E_0$ due to the interpretability of the $\lambda$ parameter. We alternately use $k$ instead of $\lambda$ to avoid confusion with the latent heat of vaporization, leading to equation (3):

$$\frac{E - kE}{E_p - kE} = \frac{Q_b}{W - kE} \tag{3}$$

In Abeshu and Li (2021), $E_0$ included interception, evaporation from surface depression, topsoil evaporation, and shallow transpiration. In Gerrits et al. (2009), they assumed that interception includes canopy and understory interception, in addition to topsoil evaporation, while deep soil evaporation is insignificant or can be combined with interception. In Savenije (2004), they considered topsoil evaporation to be a part of interception, and distinguished transpiration between fast and slow ones, where fast transpiration relies on moisture in the top 50 cm of soil, and slow transpiration relies on deeper soil moisture. Therefore, we assume that $E_0$ includes bare soil evaporation, interception, and a portion ($f$) of the transpiration ($E_t$) representing the fast transpiration from the top 10 cm of soil (Abeshu and Li, 2021; Savenije, 2004). Since root uptake not only occurs near the surface but also progresses downwards (Gardner, 1983), we assume that transpiration extracted from the topsoil occurs in a rapid manner that makes it inaccessible to the competition between baseflow and $E$, and therefore belongs to $E_0$. Therefore, $E_0$ includes all evaporative fluxes except slow transpiration, meaning that slow transpiration is the only evaporative flux that competes with baseflow. Slow transpiration can therefore be expressed as $E_{t\_slow} = E - E_0$. For transpiration, we define fast transpiration as $E_{t\_fast} = f\, E_t$, and thus slow transpiration as $E_{t\_slow} = (1 - f)\, E_t$. Equating these two $E_{t\_slow}$ equations yields $E - E_0 = (1 - f)E_t$. Substituting $E_0$ with $kE$ yields $(1 - k)E = (1 - f)E_t$, and thus we can get:

$$\frac{E_t}{E} = \frac{1 - k}{1 - f} \tag{4}$$

Equation (4) indicates that $E_t/E$ can be found using $k$ and $f$ values. The $k$ parameter can be found by applying an optimization technique that maximizes the non-parametric Kling-Gupta efficiency (KGE, equation 5, Gupta et al., 2009; Pool et al., 2018) between observed soil wetting (from watershed balance, equation 6) and simulated soil wetting (rearranging equation **(3)** to be in terms of soil wetting, equation 7).

$$KGE = 1 - \sqrt{(r - 1)^2 + (\alpha - 1)^2 + (\beta - 1)^2} \tag{5}$$

where $r$ is Pearson correlation coefficient, $\alpha$ is relative variability in the simulated and observed values, and $\beta$ is the ratio between the mean simulated and mean observed flows.

From the water balance equation at the watershed scale, we obtain observed wetting as:

$$W_{obs} = P - Q_d \tag{6}$$

And by rearranging equation (3) to obtain simulated wetting:

$$W_{sim} = Q_b \frac{E_p - kE}{E - kE} + kE \tag{7}$$

Since $f$ represents the fast response of transpiration, we follow a similar approach to Abolafia-Rosenzweig et al. (2020) in defining the ratio of surface transpiration using root distribution and soil water stress. We additionally distinguish between energy- and water-limited regions by constraining energy-limited $f$ using the aridity index as displayed in equation (8):

$$f = r_{10} \times S \times f_{AI} \tag{8}$$

Where $r_{10}$ is the root percentage in the top 10 cm of the soil, $S$ is the soil moisture availability, and $f_{AI}$ represents impact of available energy. If the aridity index (AI) is less than 1, the region is energy limited. Thus, $f_{AI} = AI$. If $AI \geq 1$, then $f_{AI} = 1$. The rationale behind this is that when $AI < 1$, only a fraction of the transpiration from the top surface layer is quantified to be part of the fast components due to its energy limited nature.

The literature shows variation in how the depth of fast transpiration is defined. For example, Abolafia-Rosenzweig et al. (2020)
used the top 5 cm to estimate transpiration from the surface soil layer. Wang et al. (2021) indicated that evapotranspiration occurs most rapidly from the top 10 cm of soil, with deeper layer responding more slowly. Similarly, Zhang et al. (2022) reported that rapid soil moisture responses to rainfall were concentrated in the top 5-10 cm, suggesting that fast transpiration is likely driven by increased soil moisture within this layer. By contrast, Abeshu and Li (2021) used 50 cm as the depth of the rapid response. We consider 50 cm to be an overestimation, as for some vegetation types (e.g., grasses) this depth may
encompass nearly the entire rooting zone. Based on this evidence, we adopted 10 cm as the representative depth for fast transpiration. In addition, we conducted a sensitivity analysis in section 4.4 to quantify the effect of this depth choice on the $E_t/E$ values.

The soil moisture availability, $S$, represents the moisture availability in the root zone for root water uptake. Abolafia-Rosenzweig et al. (2020) calculated the soil moisture availability as a function of soil moisture, wilting point, and field capacity.
To rely on hydrological observations instead of simulated or remotely sensed soil moisture, we assume the soil moisture availability to be represented by the ratio between baseflow and total streamflow ($Q_b/Q$). This ratio can give an indication of water availability in the soil and hence can be used to indicate soil moisture availability. Since we apply this method at the watershed scale, there may be multiple vegetation types in the same watershed, and therefore, we calculate a weighted value of $f$.

**2.2 Data**

From Equations 2-5 and the descriptions of Section 2.1, we see that one needs long-term observed precipitation, streamflow, baseflow, estimated $E_p$, and root distribution to estimate the $E_t/E$ ratio. Watershed boundaries and precipitation data were retrieved from the Hydrometeorological Sandbox - École de technologie supérieure (HYSETS) dataset (Arsenault et al., 2020). The HYSETS dataset includes watershed boundaries, land cover, soil properties, meteorology, and hydrological data for
14,425 watersheds in North American. We selected 648 watersheds (Figure 2) across the United States with at least 10 years of streamflow data between 1980 and 2018 from this HYSETS data source. Detailed land cover data were retrieved from the ESA CCI Land Cover project (www.esa-landcover-cci.org, last accessed December 28, 2022).

Streamflow data were retrieved from the US Geological Survey (USGS), and their corresponding baseflow magnitudes were estimated by separating it from the streamflow data using a one-parameter digital filter separation method (Lyne and Hollick,

1979). Filtering methods separate direct runoff and baseflow by differentiating them based on frequency spectrums of the hydrograph, where low frequency flow represents baseflow and high frequency represents the direct runoff which has rapid responses to precipitation. We employed the widely used filtering method tool developed by Purdue University, Web-based Hydrological Analysis Tool (WHAT, Lim et al., 2005, 2010 , https://engineering.purdue.edu/mapserve/WHAT, last accessed 25 Oct 2022), to separate baseflow from the observed streamflow.  We set the value of the filter parameter to be 0.925 which

is within the suggested range. We did a sensitivity analysis (in a separate study) and used different filter values and methods available from WHAT, the results were similar. Since other methods such as Eckhardt (2005) require knowledge of hydrogeological conditions, we chose the one-parameter digital filter method due to its simplicity and constant parameter value, which produces plausible results (Eckhardt, 2008; Xie et al., 2020). Additional details on the baseflow separation method are presented in Lim et al. (2005)

Information related to root density functions was obtained from Zeng (2001), who represented root density distribution as a two-parameter function for each vegetation type based on compiled root database. The root density distribution from Zeng (2001) was validated using root information from other studies (Fan et al., 2016; Jackson et al., 1996; Lozanova et al., 2019; Schenk and Jackson, 2002; Wallace et al., 1980). Soil moisture stress ($Q_b/Q$) was calculated based on the USGS observed streamflow and the estimated baseflow from WHAT.

Numerous $E_p$ data products are available that satisfy our study regions and time period requirements, posing a question as to which one should be selected – as each has its own strengths. To address this question, we examined six widely used $E_p$ data products and assessed their impact on the estimation of $E_t/E$ ratios. These data products were selected because they are (1) widely used within the hydrological and ecological communities, (2) associated with a wide range of spatial resolutions, and (3) derived using different methods. The six $E_p$ datasets are the Global Land Evaporation Amsterdam Model (GLEAM v3.5a,

Martens et al., 2017), the Moderate Resolution Imaging Spectroradiometer (MODIS MOD16A3GF) product (MODIS/Terra Net Evapotranspiration Gap-Filled Yearly L4 Global 500m SIN Grid V061 [Data set], 2022), the $E_p$ dataset from Zhang et al. (2010), the North American Regional Reanalysis (NARR, Mesinger et al., 2006), the Simple Process-Led Algorithms for Simulating Habitats (SPLASH v1.0, Davis et al., 2017) , and the Breathing Earth System Simulator (BESS v2, Li et al., 2023). Details of these six products are provided in Table 1.

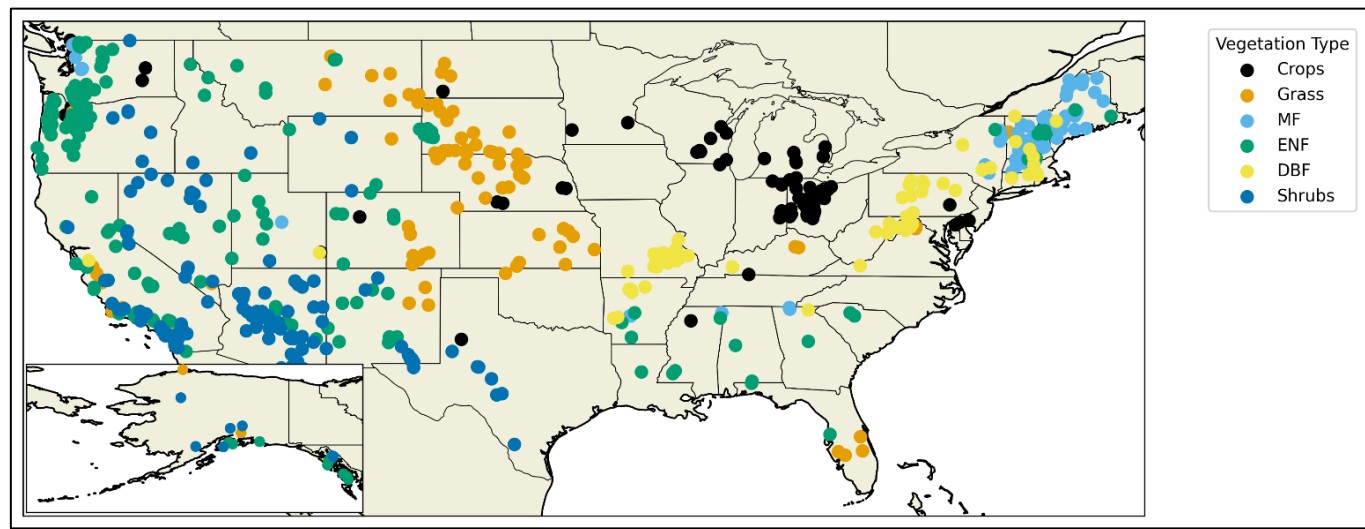

**Figure 2: 648 watersheds in the US, categorized into six vegetation types; crops, grass, shrubs, evergreen needleleaf forest (ENF), deciduous broadleaf forests (DBF), and mixed forests (MF). The inset map at the bottom left shows watersheds in Alaska.**

**Table 1: Description of six $E_p$ products used in this study.**

| Dataset | $E_p$ equation | Spatial and temporal scale | Remarks |
|---------|---------------|---------------------------|---------|
| GLEAM v3.5a | Priestley-Taylor | 0.25×0.25°, Daily/Monthly, 1980-2021 | |
| NARR | Eta Model (Penman based) | 32×32 km, Daily/Monthly, 1979-2022 | |
| MODIS MOD16A3GF | Combination of Penman-Monteith and Priestley-Taylor | 500×500m, 8-day/Yearly, 2000-2021 | |
| SPLASH | Priestly-Taylor | 1 km, Daily, 1980-2018 | Forced using daily DayMet (Thornton et al., 2022) data |
| BESS v2 | Priestly-Taylor | 5 km, Monthly, 1982-2022 | |
| Zhang | Penman-Monteith | 8×8 km, Daily/Monthly, 1983-2006 | |

Environmental variables – relative humidity, downward shortwave radiation, air temperature, wind speed, and soil moisture content – were retrieved from the NARR dataset to study the dependencies of $E_t/E$ on environmental factors. Data on leaf area

index (LAI) were obtained from the Global Monthly Mean Leaf Area Index Climatology produced by ORNL DAAC (Mao and Yan, 2019) and aggregated to obtain the long-term mean LAI at watershed scale.

The relevant data were collected for 648 watersheds and aggregated to the annual timescale. The dominant vegetation type was determined for each watershed from the ESA CCI land cover data, and watersheds were classified into six vegetation types: crops, grass, shrubs, evergreen needleleaf forest (ENF), deciduous broadleaf forest (DBF), and mixed forest (MF). We assume each watershed has a single mean long-term $E_t/E$ value. For each dataset, due to the different time coverage of the datasets and the streamflow gauges, we filtered the watersheds to include only those that have available data for at least 10 years. We used optimization to find $k$. We then performed additional filtering for each dataset to remove watersheds with KGE values less than zero. Using the filtered watersheds, we calculated $E_t/E$ based on estimated $k$ and $f$ together with the other variables. The final number of watersheds associated with each dataset used in this study, after filtering, is shown in Table 2.

Table 2: Number of filtered watersheds for each potential evapotranspiration ($E_p$) data product. Watersheds with less than 10 years of data and/or with Kling-Gupta efficiencies less than zero were removed from the analysis. Numbers are shown for each of the six vegetation types.

| Type | All watersheds | NARR | MODIS | Zhang | GLEAM v3.5a | BESS v2 | SPLASH |
|---|---|---|---|---|---|---|---|
| **Crops** | 74 | 72 | 61 | 57 | 73 | 59 | 71 |
| **Grass** | 89 | 84 | 66 | 73 | 86 | 79 | 81 |
| **Shrubs** | 146 | 131 | 107 | 114 | 134 | 128 | 131 |
| **ENF** | 206 | 166 | 118 | 118 | 173 | 161 | 156 |
| **DBF** | 65 | 65 | 61 | 54 | 65 | 64 | 65 |
| **MF** | 68 | 63 | 58 | 52 | 66 | 51 | 61 |
| **Total** | **648** | **581** | **471** | **468** | **597** | **542** | **565** |

## 3 Impact of $E_p$ products

Figure 3a shows mean annual $E_p$ values from six different data products for the 648 study watersheds. We observe large differences in mean annual $E_p$ among the six different data products. The differences in $E_p$ are likely attributed to variations in input data and parameter values used by these products, while differences in methods and resolutions used to compute $E_p$ may play a secondary role (Hassan et al., 2024). Discrepancies between the input net radiation used in different data products result in especially large variations in the computed $E_p$. Variations in parameter values, including the Priestly-Taylor α parameter, among different data products also result in significant differences in the resulting $E_p$. On the other hand, the $E/E_p$ ratios from the six different $E_p$ products are relatively consistent among the six datasets (except for GLEAM) as shown in Figure 3b. This is likely because within each product the same input/forcing data and parameter values are employed for both $E_p$ and E,

resulting in similar impacts on both. Such consistency is an indication of a uniformity of the underlying physics across these

225 five products, despite the large disparities in their individual $E_p$ magnitudes. The GLEAM $E_p$ product, which has also been previously identified for its overestimation of $E/E_p$ ratio by Peng et al. (2019) in comparison with FLUXNET $E/E_p$, appears to be an exception. Rather than excluding the GLEAM data product, we opted to adjust its $E/E_p$ ratio by normalizing it with the average ratio of the other five datasets (NARR, MODIS, Zhang, SPLASH, and BESS), yielding an adjusting factor of 0.7. This adjusting factor of 0.7 was applied to GLEAM to adjust its $E/E_p$ values. In addition, rescaled $E_p$ values from the six data

products in this study were newly derived by applying their individual $E/E_p$ ratios, obtained from their own data products, to the watershed $E$ values calculated based on watershed balance (i.e., $E = P - Q$) for each watershed. The importance of deriving $E_p$ values for each data product through this rescaling approach (referred to as rescaled $E_p$), rather than using the original $E_p$ product, is to ensure consistency between the $E_p$ values and the watershed-budget estimated $E$ values for each watershed while preserving the $E/E_p$ ratios from the individual products. This is necessary because the magnitudes of some original $E_p$ products

are smaller than their corresponding watershed-budget estimated $E$ values.

In essence, we derive new $E_p$ values for all six products using Equation (9), maintaining the $E/E_p$ ratio for each data product (except for GLEAM, whose $E/E_p$ ratio is adjusted by a factor of 0.7). This approach yields consistent $E_p$ values across the 648 watersheds for each individual data product and captures the essential variations among the six $E_p$ datasets. The rescaled $E_p$ values obtained from Equation (9) uphold the fundamental principles of individual products by preserving their respective

$E/E_p$ ratios. By doing so, the effects stemming from differences or uncertainties in their inputs/forcing data are notably mitigated, as the new $E_p$ values are calculated using the watershed-budget estimated $E$ and their own $E/E_p$ ratios. This concept is akin to the notion of emergent constraints employed by others (Green et al., 2024; Hall et al., 2019; Williamson et al., 2021).

$$E_{p_{rescaled}} = \frac{E_{p_{dataset}}}{E_{dataset}} \times E_{obs} \qquad (9)$$

where $E_{dataset}$ and $E_{p_{dataset}}$ are values extracted from different data products, and $E_{obs}$ is the watershed-budget estimated $E$ calculated as $P - Q$ based on observed $P$ and $Q$ for each watershed. Table 3 shows the correlation between the rescaled $E_p$

values of the six data products; the correlations show good consistency between the rescaled $E_p$ values. These six rescaled $E_p$ data products are then applied to Equations 2-5 to obtain $E_t/E$ ratios for each of the six vegetation types over the 648 watersheds. With the six rescaled $E_p$ data products, we can assess how variations in $E_p$ affect the robustness of our new method in estimating $E_t/E$.

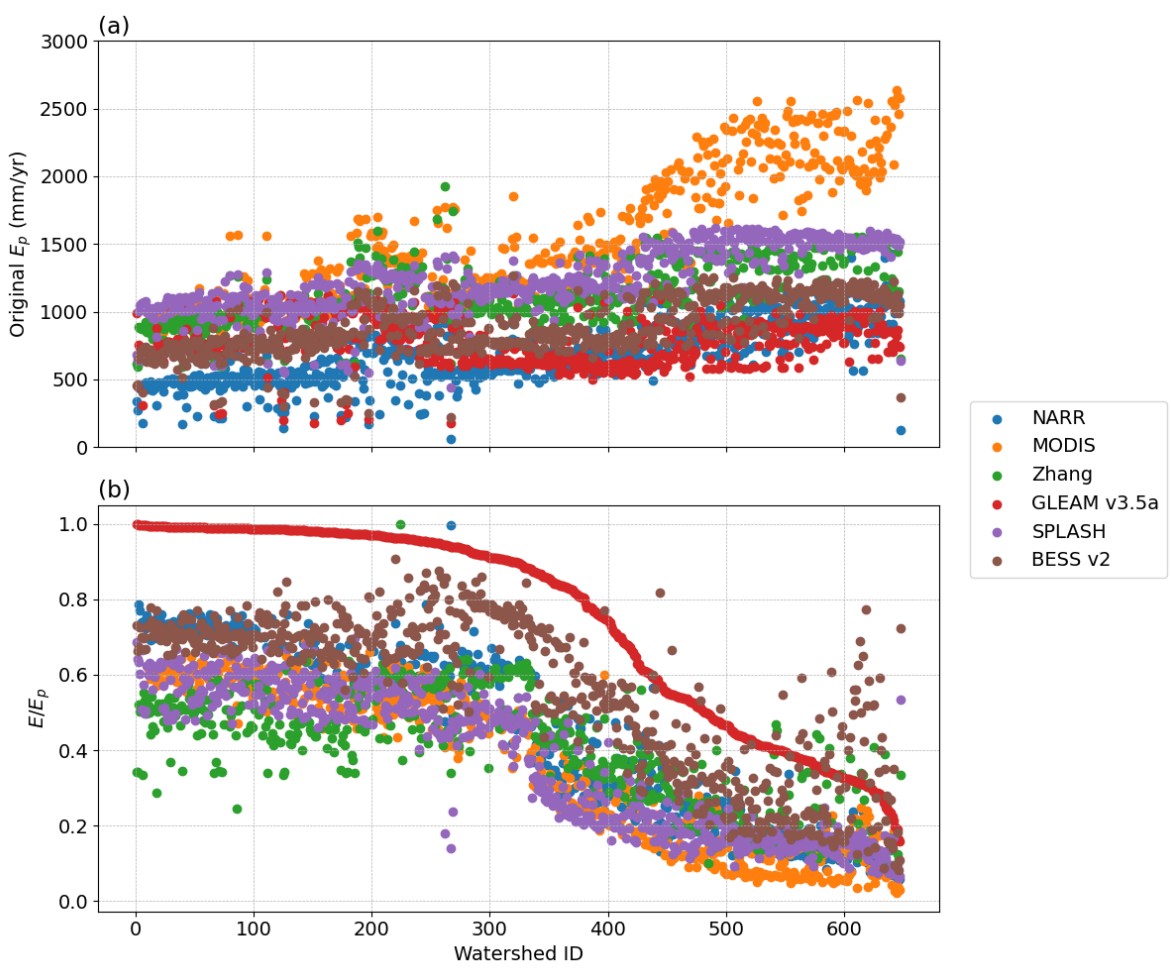

Figure 3: Original $E_p$ for six data products: NARR, MODIS, Zhang, GLEAM v3.5a, SPLASH, and BESS v2 for 648 watersheds. (a) $E_p$ values retrieved from the data products, and (b) $E/E_p$ ratios retrieved from the data products. Watersheds are sorted in descending order according to GLEAM's $E/E_p$.

Table 3: Correlations between rescaled $E_p$ of six data products: NARR, MODIS, Zhang, GLEAM v3.5a, SPLASH, and BESS v2 for 648 watersheds.

|  | *MODIS* | *GLEAM* | *NARR* | *SPLASH* | *BESS* | *Zhang* |
|---|---|---|---|---|---|---|
| MODIS | 1 |  |  |  |  |  |
| GLEAM v3.5a | 0.72 | 1 |  |  |  |  |
| NARR | 0.81 | 0.83 | 1 |  |  |  |
| SPLASH | 0.80 | 0.84 | 0.83 | 1 |  |  |
| BESS | 0.92 | 0.78 | 0.73 | 0.75 | 1 |  |
| Zhang | 0.70 | 0.83 | 0.68 | 0.69 | 0.92 | 1 |

## 4 Results

### 4.1 $k$ values

Figure 4 shows an example of the optimization between observed soil wetting ($W_{obs}$) and the simulated soil wetting ($W_{sim}$) with the optimized $k$ value for a representative watershed of each vegetation type. Figure 5 shows the estimated values of $k$ for the 648 watersheds using each of the six input datasets based on Equations 6-8. The six datasets show similar trends, where the highest $k$ values are observed for the shrubs and grass vegetation types. Crops have lower $k$ values than shrubs and grass, but equal or higher than those for forests according to the dataset used. Figure 5 illustrates that the greatest variations among the six data products occur in the mixed forest and crops. This discrepancy may be attributed to differences in how each data product defines mixed forest and crop compositions, resulting in varying estimated parameters. The $k$ values observed in our study are similar in trend to those reported by Abeshu and Li (2021), but lower in magnitude. This difference is likely due to differences in input data to the GPH equation such as precipitation and PET values since different datasets are used for both studies. Sivapalan et al. (2011) reported lower $k$ values (between 0 and 0.45). However, their definition of $k$ differs from ours: while we define $E_0 = kE$ in Eq. 3, they adopted the formulation of $E_0 = kEp$. Since actual evapotranspiration (E) is typically much smaller than potential evapotranspiration ($E_p$), it is expected that their $k$ values are lower than ours. In addition, the analysis of Sivapalan et al. (2011) was limited to 12 watersheds under relatively humid conditions (maximum aridity of 2.29), which does not capture the full range of climatic conditions, particularly drier environments.

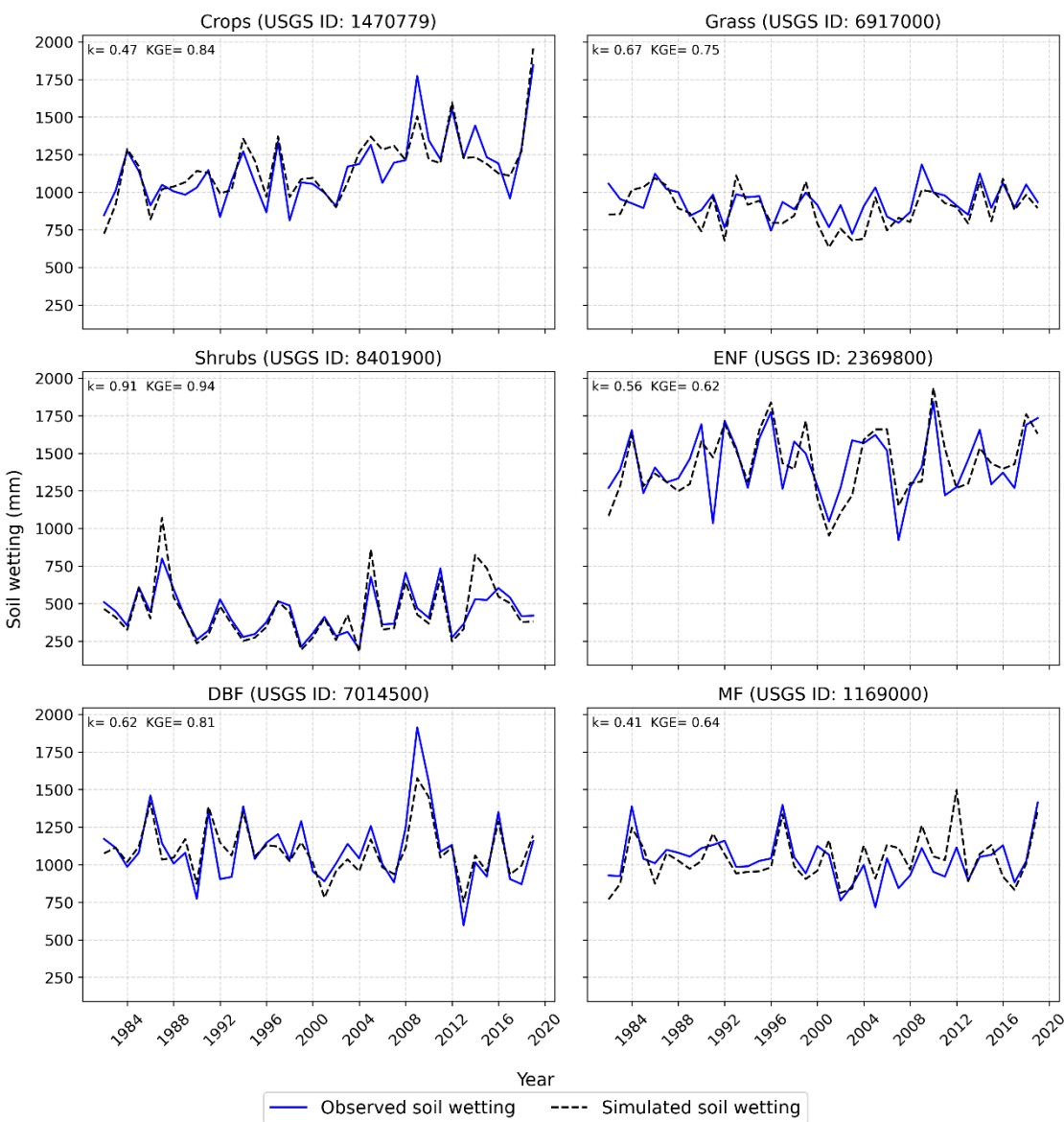

**Figure 4: Optimization of $k$ values using observed and simulated soil wetting as explained in equations 6-8. Figure shows observed and simulated soil wetting time series for an example watershed for each of the six vegetation types (crops, grass, shrubs, ENF, DBF, MF) using NARR data.**

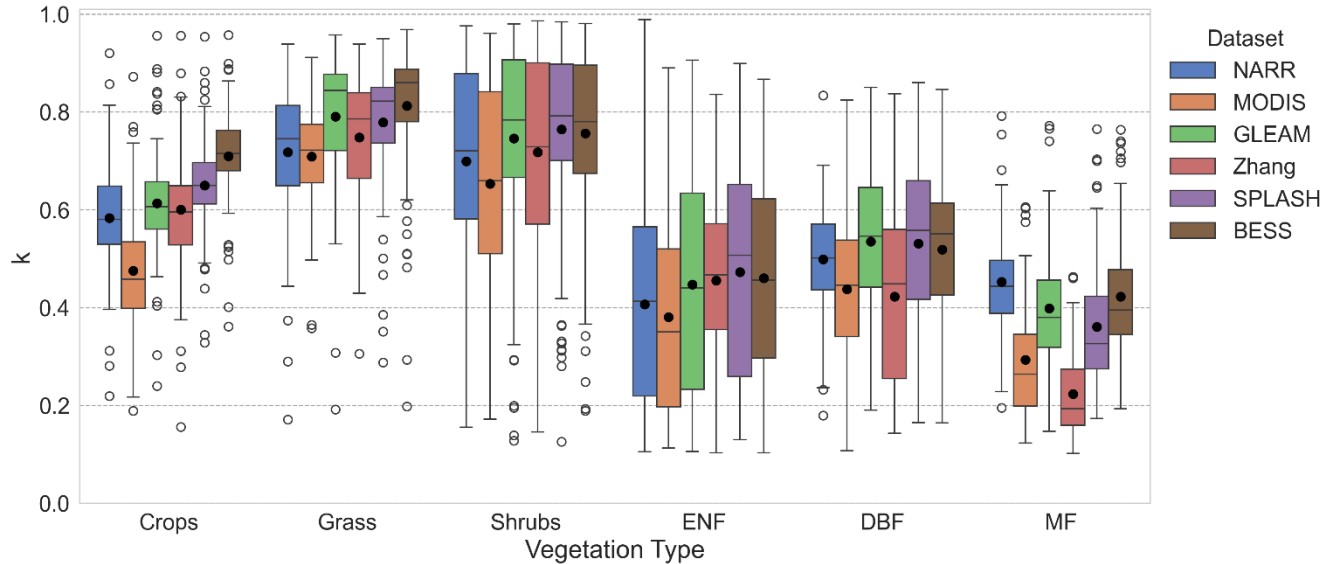

**Figure 5: $k$ values for the watersheds using data from six datasets: NARR, MODIS, Zhang et al. (2010), GLEAM after rescaling, SPLASH, and BESS. Note that ENF, DBF, and MF represent, respectively, evergreen needle-leaf forest, deciduous broadleaf forest, and mixed forest in the figure.**

### 4.2 $f$ values

Figure 6 shows the values of the $f$ parameter for 648 watersheds classified into six vegetation types. The highest $f$ value is observed in grass, which can be explained by their shallow rooting depths causing higher portions of fast transpiration. The lowest $f$ values can be observed in forests due to their deeper rooting system, which provides access to deeper soil moisture, reducing the portion of fast transpiration.

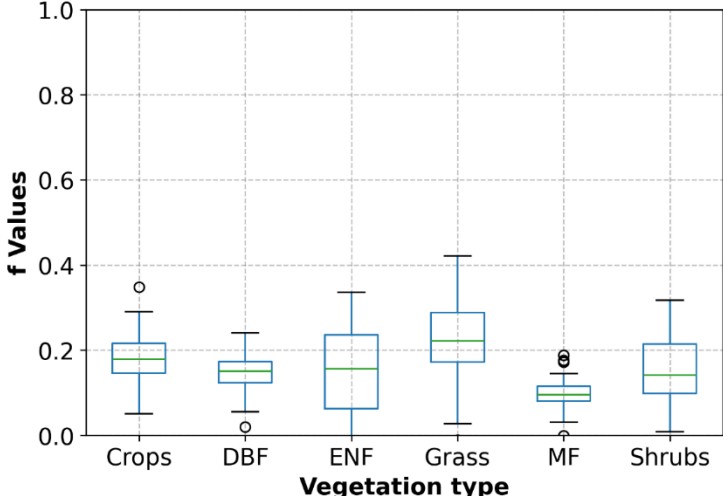

**Figure 6: $f$ values for six vegetation types for 648 watersheds**

## 4.3 Et/E values

$E_t/E$ ratios are shown in Figure 7 and Table 4. Overall, the trend is consistent among the six datasets. Grass and shrubs have the lowest $E_t/E$ values, with mean $E_t/E$ in the range of 0.19-0.39. Crops have higher mean $E_t/E$ ratios, with NARR, Zhang, and GLEAM averaging around 0.4, while MODIS and SPLASH show a higher crop mean $E_t/E$ of around 0.51. BESS has the lowest crop $E_t/E$ with a value of 0.29. All datasets have similar forest $E_t/E$ trend, with lowest mean $E_t/E$ for DBF (0.46-0.60), followed by ENF (0.52-0.71). The highest mean $E_t/E$ is exhibited for MF (0.55-0.76).

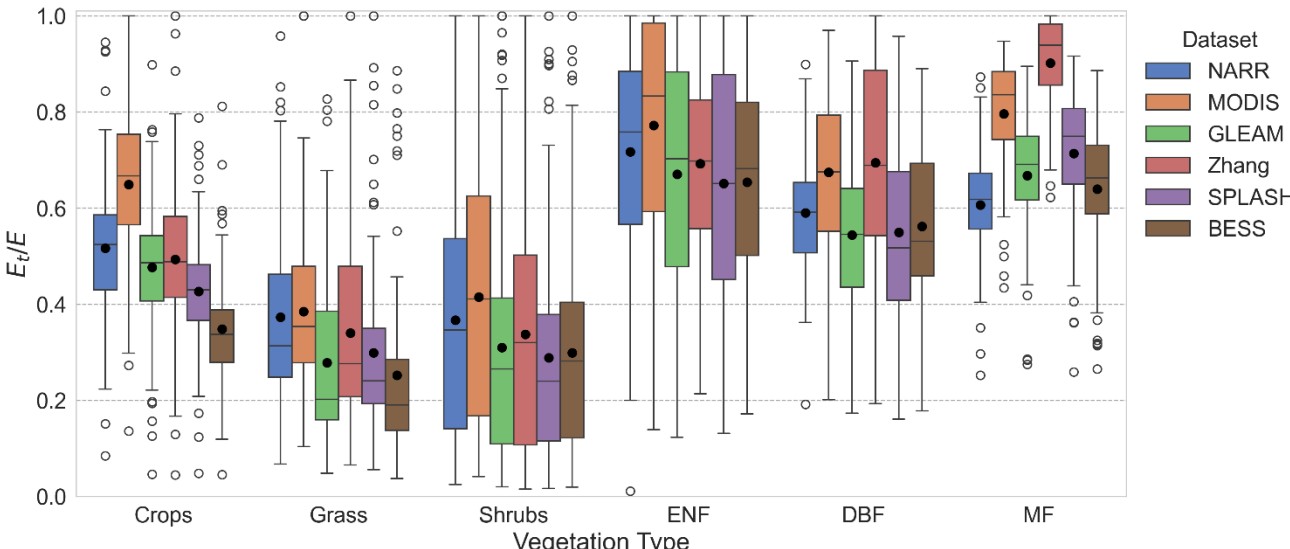

**Figure 7: E_t/E values for the watersheds using data from the six datasets: NARR, MODIS, Zhang et al. (2010), GLEAM after rescaling, SPLASH, and BESS**

**Table 4: Mean E_t/E values for six vegetation types using E_p data from the six data products. Minimum, maximum, and mean values are shown for each vegetation type.**

| Data product | Crops | Grass | Shrubs | ENF | DBF | MF | Mean |
|---|---|---|---|---|---|---|---|
| **NARR** | 0.52 | 0.37 | 0.37 | 0.72 | 0.59 | 0.61 | 0.52 |
| **MODIS** | 0.65 | 0.38 | 0.41 | 0.77 | 0.67 | 0.80 | 0.59 |
| **Zhang** | 0.49 | 0.34 | 0.34 | 0.69 | 0.69 | 0.90 | 0.52 |
| **GLEAM** | 0.48 | 0.28 | 0.31 | 0.67 | 0.54 | 0.67 | 0.48 |
| **SPLASH** | 0.43 | 0.30 | 0.29 | 0.65 | 0.55 | 0.71 | 0.47 |
| **BESS** | 0.35 | 0.25 | 0.30 | 0.65 | 0.56 | 0.64 | 0.45 |
| **Minimum** | **0.35** | **0.25** | **0.29** | **0.65** | **0.54** | **0.61** | **0.45** |
| **Maximum** | **0.65** | **0.38** | **0.41** | **0.77** | **0.69** | **0.90** | **0.59** |

| | | | | | | | |
|---|---|---|---|---|---|---|---|
| **Mean** | 0.48 | 0.32 | 0.33 | 0.69 | 0.60 | 0.70 | 0.50 |

**4.4 Sensitivity of $E_t$/E to $f$ values**

We perform a sensitivity analysis to investigate the effect of soil depth used in estimating $f$ on the $E_t$/E values. Since $f =$
$r_{10} \times S \times f_{AI}$, and both $S$ and $f_{AI}$ are constant for the watershed, differences in $f$ arise from changes in $r_{10}$. Therefore, we tested the effect of using different depths of rapid response (5 cm, 10 cm, and 15 cm) on the resulting $E_t$/E values, which are shown in Figure 8. We selected 5 and 10 cm based on the general consensus in the literature and extended the range to 15 cm to account for additional uncertainty. These depths represent plausible values for fast transpiration, and as discussed in Section 2.1, we do not consider larger depths to contribute significantly as fast transpiration.

The percentage and absolute changes in $E_t$/E resulting from variations in rapid response depth are summarized in Table 5 as average change per vegetation type (with six data products averaged for each type). The full results for individual data products are provided in Appendix A (Tables A1-A6). The largest percentage changes were observed for the grass type, with $E_t$/E varying by about 10-13% when the depth was increased or decreased by 5 cm from the 10 cm reference. The largest absolute change occurred when the depth was increased from 5 cm to 15 cm for the ENF vegetation type, with a difference of 0.108.

Overall, the differences due to changing the fast response depth are minor and remain well within the uncertainty ranges reported in the literature for evapotranspiration partitioning methods, as noted in the introduction.

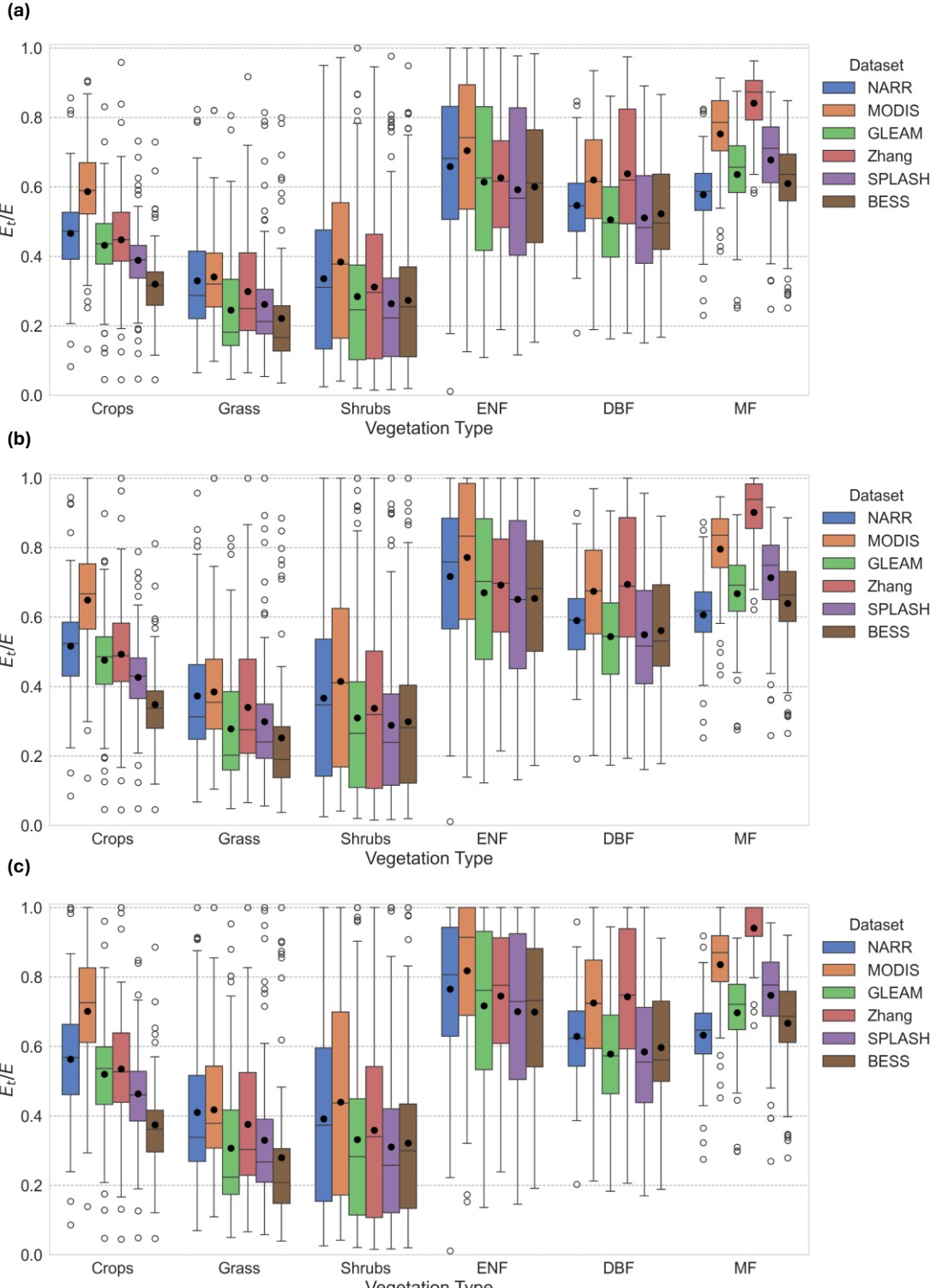

**Figure 8: Sensitivity of Et/E to different depths of fast transpiration responses: (a) 5 cm; (b) 10 cm; and (c) 15 cm.**

**Table 5: Relative and absolute change in mean $E_t/E$ values due to changes in fast transpiration depth. Results are shown as an average of the change in the six data products for each vegetation type.**

| Type | % Change in Et/E | | | Absolute change in Et/E | | |
|---|---|---|---|---|---|---|
| | 5 cm to 10 cm relative to 5 cm | 10 cm to 15 cm relative to 10 cm | 5 cm to 15 cm relative to 5 cm | 5 cm to 10 cm | 10 cm to 15 cm | 5 cm to 15 cm |
| Crops | 10.04 | 8.50 | 19.40 | 0.045 | 0.041 | 0.086 |
| Grass | 13.58 | 10.09 | 25.05 | 0.038 | 0.032 | 0.070 |
| Shrubs | 8.93 | 6.86 | 16.41 | 0.027 | 0.023 | 0.050 |
| ENF | 9.47 | 6.98 | 17.12 | 0.060 | 0.048 | 0.108 |
| DBF | 7.98 | 6.72 | 15.24 | 0.045 | 0.041 | 0.085 |
| MF | 5.49 | 4.52 | 10.26 | 0.038 | 0.033 | 0.071 |

## 5 Discussion

### 5.1 $k$ and $E_t/E$ ratios

Shrubs and grass showed higher $k$ values, likely due to their occurrence in arid and semi-arid regions in the US. The high $k$ values could be explained by the higher bare soil evaporation expected in arid regions (Baver et al., 1972), especially due to the sparse nature of shrubs, increasing bare areas and thus bare soil evaporation (Liu et al., 2022). Also, the high aridity is expected to cause water stress, lowering the continuing transpiration (portion of transpiration not included in $k$). The lower $k$ values in crops and forests may be due to the higher vegetation coverage in these areas which provides shade to the soil, reducing the amount of soil evaporation (Baver et al., 1972). Additionally, litter contributes to reducing soil evaporation and may even have a larger reduction effect than canopy shade (Magliano et al., 2017). The broader leaves of DBF increase their interception compared to ENF, thus resulting in a higher $k$ value as well.

These estimated mean $E_t/E$ ratios followed explainable trends, with shrubs and grass watersheds showing low $E_t/E$ ratios, forests exhibiting higher $E_t/E$ ratios, and crops falling in between. Given greater water availability in crops and forests, it is expected that they would exhibit higher $E_t/E$ ratios. Many crops in the US benefit from continuous irrigation, reducing water stress and promoting transpiration. Forests, with their dense canopy cover offering shade, reduce soil evaporation (Baver et al., 1972) and consequently boost the $E_t/E$ ratios. Crops also show high vegetation coverage, thereby providing shade to the soil and increasing $E_t/E$ (Baver et al., 1972). Moreover, in arid regions dominated by shrubs, lower soil water content is anticipated, resulting in diminished root water uptake (Gardner, 1983). Furthermore, the shedding of leaves in deciduous forests reduces transpiration when examined over the whole year (as here), resulting in a decreased $E_t/E$ ratio for DBF.

Differences in study scale may hinder the comparison with other studies, since our method estimates $E_t/E$ at the watershed scale, while other studies are based at a plot-scale (field/eddy covariance-based methods) or grid scale (models and remote-sensing methods). Factors affecting watershed scale $E_t/E$ include the possible presence of secondary vegetation within the watershed and the possible sparseness of the primary vegetation and presence of bare areas which can increase soil evaporation and reduce $E_t/E$, especially for shrublands. Therefore, this method has the advantage of providing a realistic watershed $E_t/E$ ratio that accounts for multiple vegetation types and sparseness in vegetation distribution. Consistent results across different datasets underscore the reliability of our new method, irrespective of the data product employed (see Fig. 5 and Table 3).

## 5.2 Effect of hydrological indices on $E_t/E$

We explore the sensitivity of $E_t/E$ to two hydrological indices, namely the runoff ratio (Q/P) and the baseflow ratio ($Q_b/Q$). Figure 9a shows a proportional relationship between $E_t/E$ and Q/P. The relationship appears to manifest as two distinct linear correlations, with arid catchments showing a steeper slope than humid catchments. Arid regions typically experience minimum runoff as a significant portion of precipitation evaporates in various forms owing to elevated atmospheric demand. This phenomenon yields high $E_t/E$ ratios at relatively low Q/P values. Conversely, humid catchments often experience substantial runoff, attributed to either saturation excess or infiltration excess runoff mechanisms, resulting in elevated Q/P ratios compared to arid catchments at equivalent $E_t/E$ values. In both cases, a higher Q/P ratio signifies increased water availability, consequently leading to higher $E_t/E$ ratios.

In Figure 9b a non-linear positive relationship is depicted between the mean $E_t/E$ and Qb/Q (baseflow ratio). The baseflow ratio serves as an indicator of soil water availability, as higher baseflow typically corresponds to increased soil moisture content (Hurkmans et al., 2008). Consequently, a positive correlation between $E_t/E$ and the baseflow ratio is anticipated. Notably, the majority of arid catchments cluster in the low Qb/Q and low $E_t/E$ region, while transitioning toward wetter catchments naturally augments both Qb/Q and $E_t/E$.

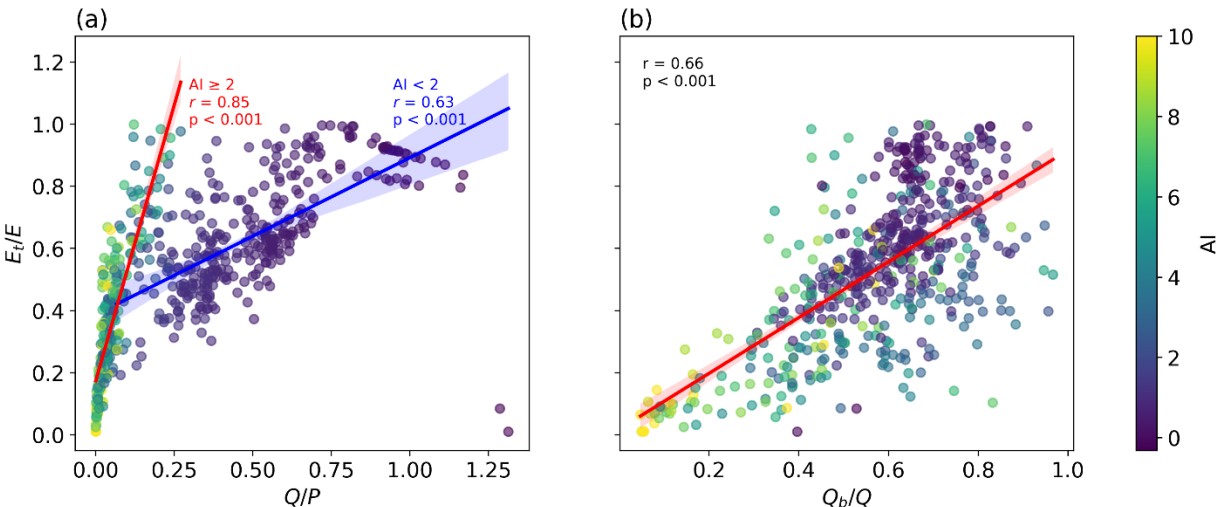

**Figure 9: Relationship between mean E$_t$/E and two hydrological indices (a) Q/P and (b) Qb/Q for 648 watersheds based on NARR data. Plots are colored according to aridity index.**

### 5.3 Effect of LAI on E$_t$/E

The leaf area index (LAI), representing the leaf area per unit ground area, reflects the combined influences of leaf size and canopy density. As shown in Figure 10, LAI appears to exert some influence over evapotranspiration partitioning. Arid watersheds show lower LAI values, and E$_t$/E ratios increase non-linearly with LAI. However, as watersheds transition toward higher humidity levels, their LAI and E$_t$/E ratios increase non-linearly, albeit at different rates. In arid regions, plants tend to reduce their leaf area to mitigate water loss (Chaves et al., 2003) decreasing both LAI and E$_t$/E – a direct consequence of high

aridity. This suggests that aridity plays a role in regulating E$_t$/E. Figure **10** illustrates a complex relationship between LAI and E$_t$/E, characterized by substantial scatter. Our findings align with previous studies indicating diverse dependence of E$_t$/E on LAI. For instance, LAI has been shown to provide a control on E partitioning (Li et al., 2019; Wang et al., 2014; Wei et al., 2017), but that effect varies from one study to another. Wang et al. (2014) showed that LAI has a non-linear relationship with E$_t$/E during the growing season, whereas Li et al. (2019) showed a weak linear relationship between mean growing season LAI and mean annual E$_t$/E across sites, with the E$_t$/E and LAI relationship within the same site being non-linear. Additionally, Cao

et al. (2022) showed a non-linear positive relationship between annual E$_t$/E and LAI.

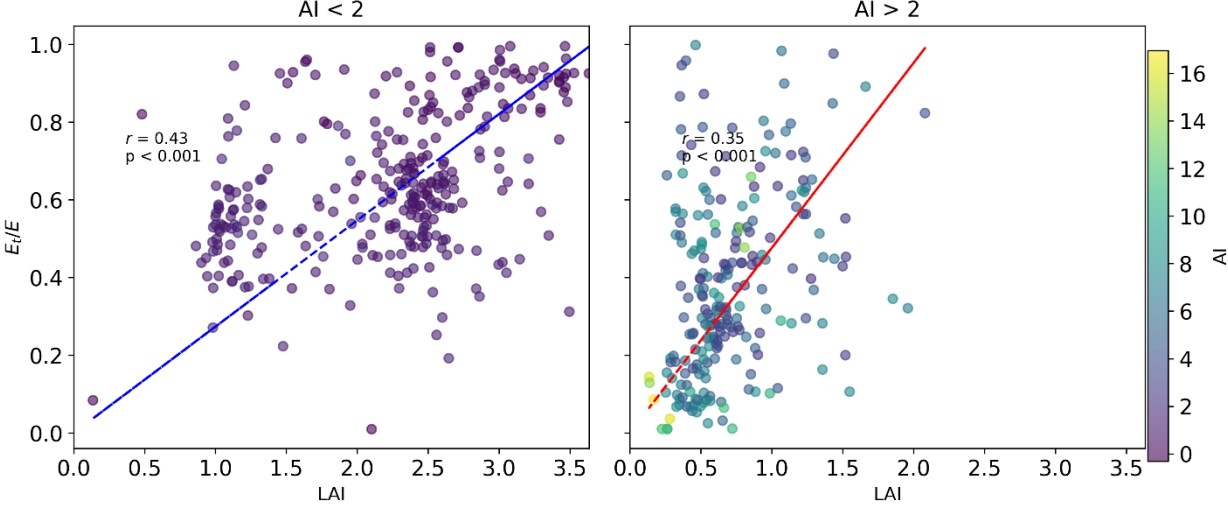

**Figure 10: Relationship between E$_t$/E and LAI for 648 watersheds using E$_t$/E calculated based on the NARR dataset.**

### 5.4 Impacts of environmental variables on E$_t$/E ratios

We explore the effect of six environmental factors on the mean E$_t$/E ratios. They are aridity index (AI), relative humidity (RH), air temperature (T$_{air}$), downward shortwave radiation (DSW), soil moisture, and wind speed (WS). These factors were derived from the NARR dataset, and the E$_t$/E ratios were calculated based on the same dataset. Since some of these environmental variables are highly correlated (as shown in Figure 11), we first perform variable selection using stepwise regression and Lasso regression to identify those that are strongly correlated with each other. Stepwise regression aims to select a subset of variables

that provide the best prediction with minimum redundancy, while Lasso regression adds a penalty term to reduce the coefficients of insignificant variables. Both methods resulted in the elimination of downward shortwave radiation, while stepwise selection additionally eliminated relative humidity and air temperature. Table 6 shows the coefficients of the environmental variables and their significance for both stepwise and Lasso regression. Although the significance test shows that air temperature and relative humidity have an insignificant impact on the Lasso regression, while the aridity index, soil

moisture, and wind speed are significant (Table 5), they are still included because they marginally contribute to the model's predictive power. Additionally, they represent independent and observable dimensions, distinct from the other three significant environmental variables.

A negative non-linear correlation between E$_t$/E and AI is present. Increased aridity prompts plants to adopt water conserving strategies (Chaves et al., 2003), thereby reducing the transpiration ratios. In humid regions, the relationship between E$_t$/E and

AI is more discernible, with AI accounting for a significant portion of the variance of E$_t$/E. Conversely, for arid regions, particularly those dominated by shrubs, the relationship shows greater scatter, suggesting that AI exerts a relatively smaller effect on E$_t$/E, while other factors play a more prominent role. Furthermore, higher air temperature contributes to lowering E$_t$/E (see Figure 12b), as it prompts water-conserving behaviors in plants and elevates soil evaporation, consequently reducing

$E_t/E$ ratios. Conversely, increasing soil moisture leads to enhanced water availability for plant root uptake, resulting in a near

linear increase in $E_t/E$, as shown in Figure 12c. The relationship between wind speed (WS) and $E_t/E$ is inconclusive; this finding is consistent with several previous studies (Dixon and Grace, 1984; Huang et al., 2015; Schymanski and Or, 2016) which have presented a mixed effect of wind speed on transpiration. Nevertheless, the effects of other environmental variables on $E_t/E$ demonstrate explainable patterns as discussed here. The other five data products (MODIS, Zhang, GLEAM, SPLASH, and BESS) show similar impacts of all the environmental variables on $E_t/E$ as those shown in Figure 12 for NARR.

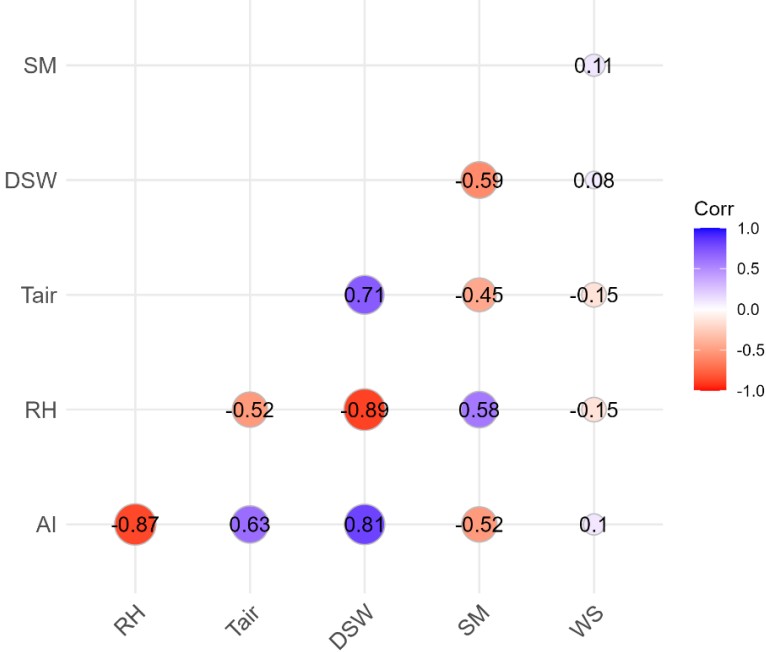

**Figure 11: Correlation between environmental variables. AI: aridity index, RH: relative humidity, Ta: air temperature, DSW: downward shortwave radiation, SM: soil moisture, WS: wind speed.**

**Table 6: Coefficients of standardized environmental variables regressed against $E_t/E$ using stepwise selection and Lasso regression. Significance levels are shown next to the coefficients (\*\*\*: p<0.001, \*\*: p<0.01, \*: p<0.05, blank: p>0.1**

|  | Coefficient (Stepwise selection) | Coefficient (Lasso regression) |
|---|---|---|
| AI | -0.105*** | -0.026*** |
| RH |  | 0.001 |
| Tair |  | -0.004 |
| DSW |  |  |
| SM | 0.066*** | 0.0005*** |
| WS | 0.023** | 0.037* |

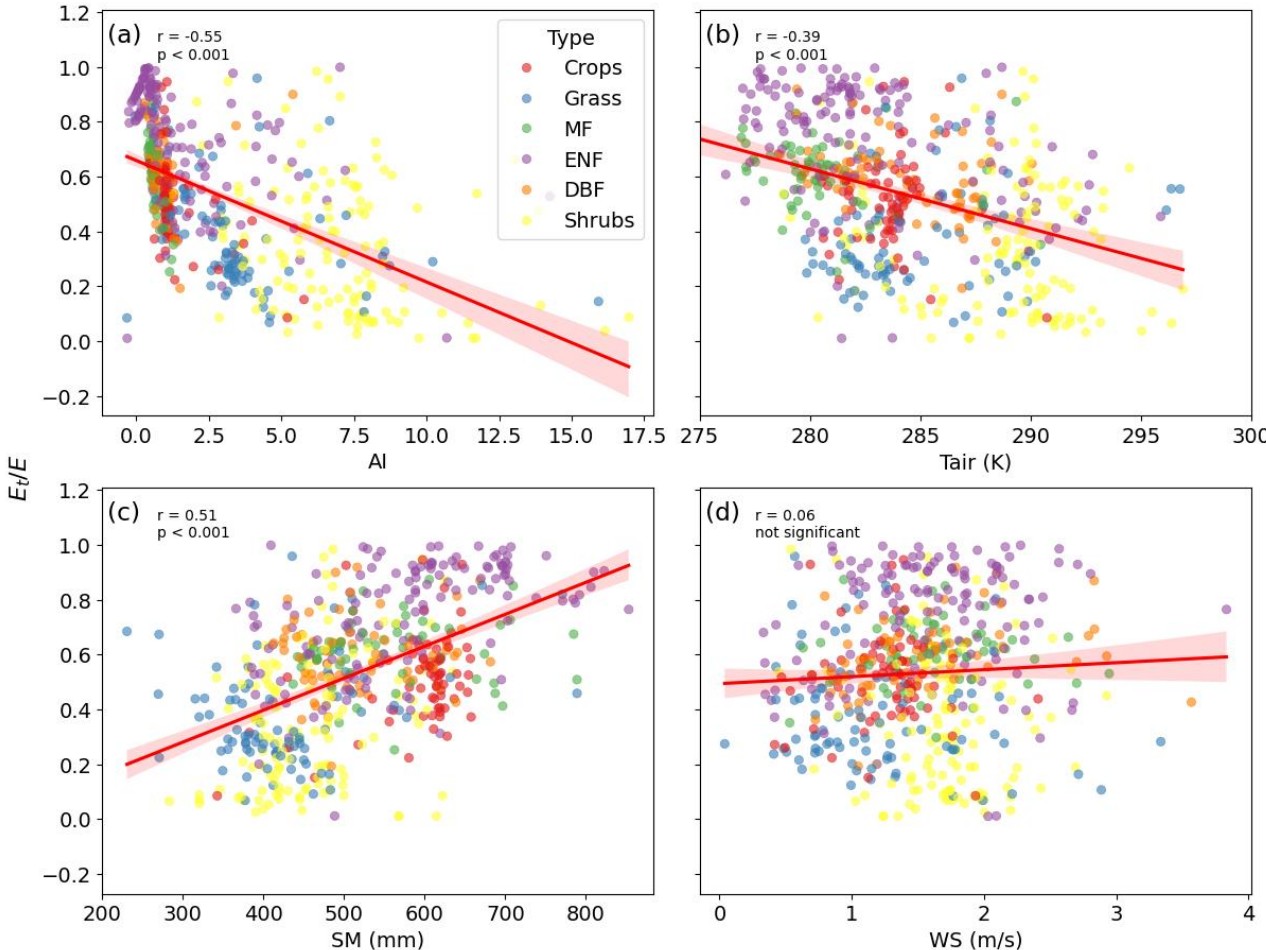

**Figure 12: Relationships between mean annual E$_t$/E and environmental factors (a) aridity index (E$_p$/P), (b) air temperature (Tair), (c) soil moisture (SM), and (d) wind speed (WS) for 648 watersheds. E$_t$/E is calculated based on NARR data, and the environmental variables are also retrieved from the NARR product. Significance of the pairwise relationships between E$_t$/E and the environmental variables are shown on each plot.**

## 5.5 E$_t$/P ratios

We computed transpiration to precipitation (E$_t$/P) ratios based on E$_t$/E values calculated from the six adjusted E$_p$ data products. The mean E$_t$/P ratios from these six datasets range from 0.24 to 0.36, aligning closely with the global mean E$_t$/P of 0.39 estimated by Schlesinger and Jasechko (2014).

We also compared our estimated E$_t$/P ratios to the E$_t$/P versus aridity index relationship identified by Good et al. (2017). Good et al. (2017) presented this relationship based on a compilation of field studies, three remote-sensing based models, and an ecohydrological model, revealing good consistency among the various E$_t$/P data sources. Figure 13 shows a similar trend to

that presented in Fig. 1 of Good et al. (2017), with the maximum $E_t/P$ ratio close to the intersection between water and energy-limited states. This maximum $E_t/P$ corresponds to an aridity index ranging between 2 and 3 in our study, similar to the estimated aridity index range of 1.3 to 1.9 for the maximum $E_t/P$ as reported by Good et al. (2017). Moreover, the maximum $E_t/P$ shown in Figure **13** ranges between 0.5 and 0.58, consistent with the maximum $E_t/P$ of 0.6 based on field data in Good et al. (2017). Notably, there is greater variation on the right side of the curve (indicating more arid conditions) compared to the left side (representing wetter conditions). In arid regions, transpiration is influenced not only by aridity, but also by factors such as groundwater table depth and soil moisture content, resulting in higher variability in the $E_t/P$ versus aridity index (AI) relationship. The consistency between Good et al. (2017) and this study suggests that this relationship holds not only at the field and remote sensing scales (as shown by Good et al., 2017), but also at the watershed scale, as demonstrated in this study. This relationship holds significance for studies like that of Cai et al. (2023) and Zhou et al. (2025) where $E_t/P$ serves as a parameter (referred to as $f_0$ in their study) to determine water-limited fAPAR and LAI. Cai et al. (2023) estimated $E_t/P$ as a global mean using non-linear regression, with a value of 0.62, akin to the maximum $E_t/P$ of 0.5 to 0.58 estimated by our fitted curves depicted in Figure **13**. Zhou et al. (2025) used a variable $E_t/P$ as a function of AI, akin to our fitted curves. Their maximum $E_t/P$ of 0.65 occurred at an AI of 1.9, similar to our fitted curves.

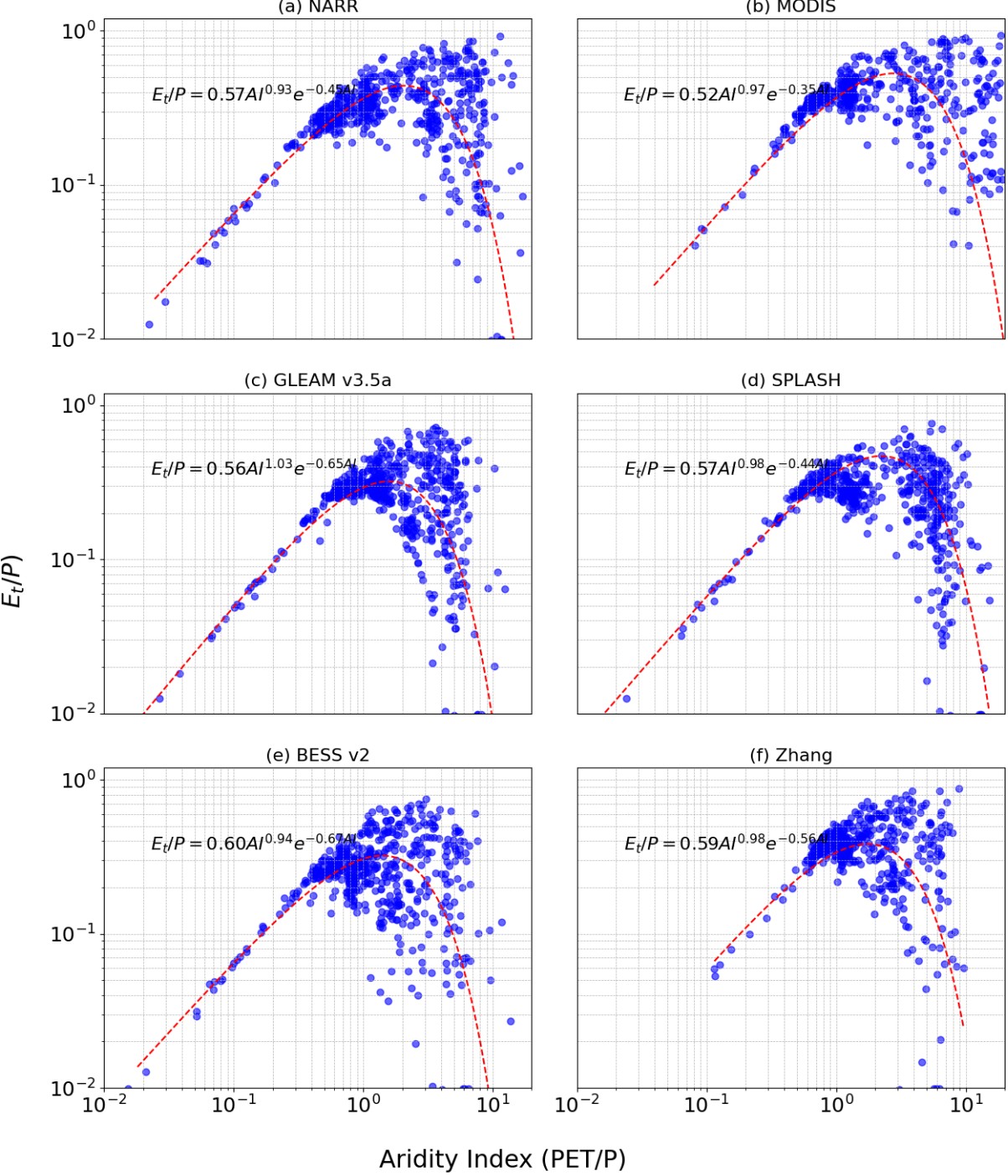

**Figure 13: E_t/P versus the aridity index for six datasets: (a) NARR, (b) MODIS, (c) Zhang et al. (2010), (d) GLEAM after rescaling, (e) SPLASH, (f) BESS**

## 6 Variation of evapotranspiration partitioning methods

Figure 7 demonstrates the influence of the six adjusted $E_p$ data products on the $E_t/E$ ratios by our new method for each vegetation type, while Table 4 provides their variation range between the minimum and maximum mean $E_t/E$ ratios. On the
440 other hand, as outlined in the introduction, estimated global mean values of $E_t/E$ from various existing methods exhibit a considerable variation, ranging from 0.24 to 0.9 (Liu et al., 2022; Wei et al., 2017). This variation may be attributed to several factors, including data inconsistencies, geographical disparities, and differences in selected time periods, apart from differences in methodology. In an effort to explore what may be the cause for the large variation among the different methods, we have tried to mitigate these factors by using the same half-hourly eddy covariance data from the FLUXNET and AMERIFLUX
ONEFLUX towers measurements in the US for the same locations and same time periods. Such an approach would allow us to elucidate the disparities among the existing E partitioning methods, consequently, providing insights on influences by different $E_p$ datasets in our method versus current existing different methods on the large range of $E_t/E$ ratios.

The four methods we selected to investigate are: (1) Zhou et al. (2016), (2) Scott and Biederman (2017), (3) Li et al. (2019), and (4) Yu et al. (2022). These four methods are selected because they are based on eddy covariance measurements whose
data are widely available, unlike sap flow and isotope measurements. Since these methods are based on flux measurements, they can be considered as field-based estimations of $E_t/E$. We apply these four methods to the same datasets from the FLUXNET and AMERIFLUX ONEFLUX towers in the US, but the final number of flux towers included for each method depends on the filtering criteria in each method and the limitations in applying each method.

The first method by Zhou et al. (2016) is based on the water use efficiency. The ratio $E_t/E$ is estimated as the ratio between the
455 apparent water use efficiency ($WUE_a = GPP \times \frac{VPD^{0.5}}{ET}$) and the potential water use efficiency ($WUE_p = GPP \times \frac{VPD^{0.5}}{T}$). Assuming that $E_t/E$ approaches 1 at some time during the growing season, the $WUE_p$ is estimated from the 95[th] quantile regression of the half-hourly scatter plot (based on all half-hourly data for the site) between $GPP \times VPD^{0.5}$ and E and is assumed to be constant for the flux tower. $WUE_a$ is then estimated for each time step as the linear regression of the E and $GPP \times VPD^{0.5}$ relationship using half-hourly data for the desired time period, which can be 8-day, monthly or annually.

The second method by Scott and Biederman (2017) is based on water use efficiency to estimate multiyear monthly average $E_t/E$ ratios. This approach estimates transpiration as the product of the inverse of the marginal water use efficiency, the ratio between transpiration WUE and marginal WUE, and GPP. The inverse of the marginal WUE is estimated from the linear regression of the GPP versus E scatter plot. The ratio between transpirational and marginal WUEs is assumed to be 1. This method requires multiple years of data for its application.

The third method by Li et al. (2019) is based on the stomatal conductance model of Lin et al. (2018) to partition evapotranspiration. The $E_t/E$ ratio is equivalent to the ratio between canopy conductance and ecosystem conductance. The eddy covariance data are divided into soil moisture bins to calibrate the parameters. Therefore, the method requires soil moisture data, along with GPP, VPD, E, and three calibrated parameters to estimate the $E_t/E$ ratio.

The fourth method by Yu et al. (2022) combines the water use efficiency with the Medlyn et al. (2011) stomatal conductance
model. This method relies on GPP, E, $C_a$, $P_a$, and VPD from the flux tower data in addition to the parameter $g_1$ from the
Medlyn et al. (2011) model. The authors compared their method to other methods and showed a high correlation with the Zhou
et al. (2016) but a low correlation with the Li et al. (2019) method.

Additionally, we compare our results to $E_t/E$ values for 20 global flux towers from Tan et al. (2021). $E_t/E$ was calculated based
on flux tower data and P-model (Stocker et al., 2020; Wang et al., 2017) outputs.

The estimated $E_t/E$ ratios from the five methods are shown in Figure 14a – e and Table 4, respectively, for the same six different
vegetation types as shown in Figure **7** with our new method.

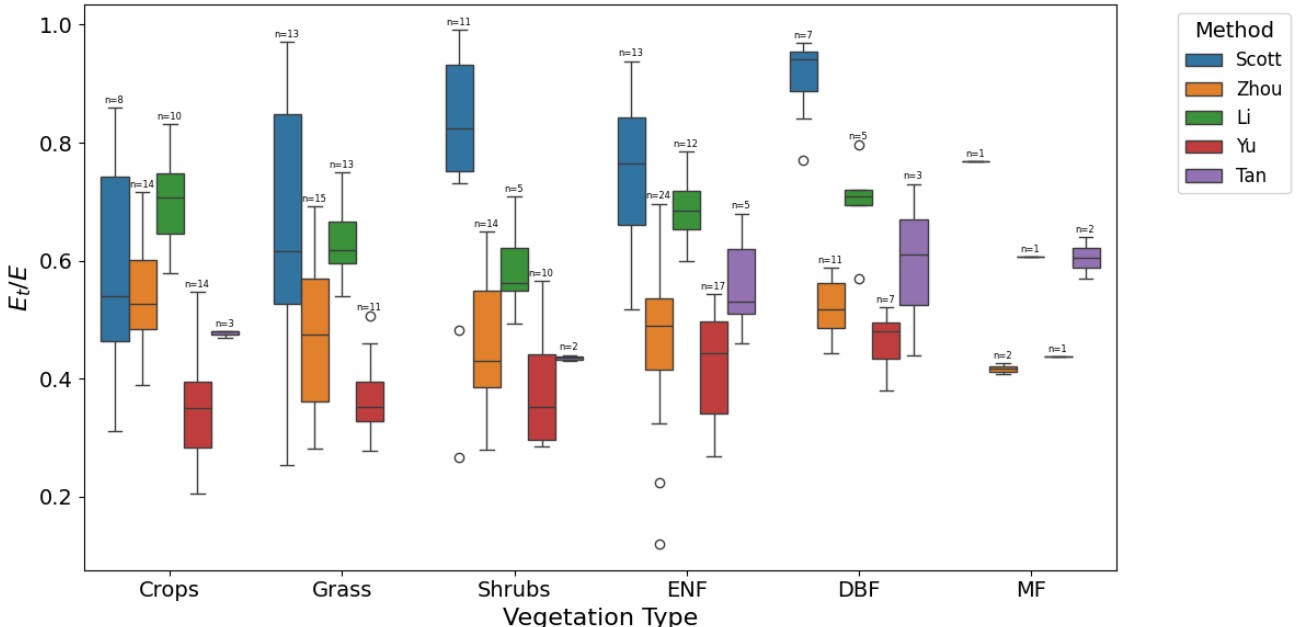

**Figure 14: $E_t/E$ values based on the eddy covariance tower data with 5 methods: (a) Zhou et al. (2016) (n=80), (b) Scott and
Biederman (2017) (n=53), (c) Li et al. (2019) (n=46), (d) Yu et al. (2022) (n=60) (e) Tan et al. (2021) (n=15).**

**Table 7: Mean $E_t/E$ values for six vegetation types using four evapotranspiration partitioning methods. Minimum, maximum, and
mean values are shown for each vegetation type.**

| Evapotranspiration partitioning method | Crops | Grass | Shrubs | ENF | DBF | MF | Mean |
|---|---|---|---|---|---|---|---|
| Zhou et al. (2016) | 0.54 | 0.48 | 0.46 | 0.46 | 0.52 | 0.42 | 0.48 |
| Scott and Biederman (2017) | 0.56 | 0.59 | 0.65 | 0.66 | 0.65 | 0.77 | 0.62 |
| Li et al. (2019) | 0.70 | 0.63 | 0.59 | 0.69 | 0.70 | 0.61 | 0.66 |

| | | | | | | | |
|---|---|---|---|---|---|---|---|
| **Yu et al. (2022)** | 0.34 | 0.37 | 0.38 | 0.43 | 0.46 | 0.44 | 0.39 |
| **Tan et al. (2021)** | 0.48 | - | 0.44 | 0.56 | 0.6 | 0.61 | 0.54 |
| **Minimum** | 0.34 | 0.37 | 0.38 | 0.43 | 0.46 | 0.42 | 0.39 |
| **Maximum** | 0.70 | 0.63 | 0.65 | 0.69 | 0.70 | 0.77 | 0.66 |
| **Mean** | 0.52 | 0.52 | 0.50 | 0.56 | 0.59 | 0.57 | 0.54 |

The inconsistencies among the five methods are evident, with Zhou, Yu, Li, and Tan showing minimal variation among vegetation types, while Scott displays substantial variation. Moreover, the magnitudes and trends of $E_t/E$ across these methods are also inconsistent. These discrepancies indicate a lack of agreement on both the mean $E_t/E$ values and the variation ranges among the different methods. Consequently, these methods are not suitable as reference points for evaluating our new method. Instead, the assessment of our new method should be based on its physical behavior and relationships with other variables, as discussed in Section 5. It is noteworthy that compared to Figure 7, the variation range of $E_t/E$ ratios from the five different methods, utilizing the same data at the same locations, is significantly greater than that for our new method in which disparity is attributed to the variations associated with the $E_p$ methods employed. Additionally, since our method is at a larger (watershed) scale, we observe larger variations between vegetation types, which can be attributed to different vegetation densities and bare land percentages at larger scales which is not a factor at smaller (flux tower) scales.

## 7 Conclusions

We have presented a new method for determining the transpiration to total evapotranspiration ($E_t/E$) ratio using long-term hydrological observations. This method is based on the generalized proportionality hypothesis, which has wide applications in hydrology. We applied the method to 648 watersheds in the US using six different $E_p$ data products. Our findings demonstrate consistent $E_t/E$ results across these diverse $E_p$ datasets, facilitated by a rescaling of $E_p$ derived from the $E/E_p$ ratios obtained from each individual data product and watershed-budget estimated E computed from the watershed water balances.

Our analysis reveals that varying $E_t/E$ ratios across watersheds are associated with different vegetation types, with shrubs and grasslands exhibiting lower $E_t/E$ values compared to crops and forests. Furthermore, our results underscore the significant influence of leaf area index (LAI), hydrological indices (Q/P and Qb/Q), and prevailing environmental conditions on $E_t/E$. Our method also provides a realistic estimate of $E_t/E$ at a watershed scale that implicitly accounts for the heterogeneity of vegetation within the catchment. Our method can also be useful for constraining hydrological models, land surface models, and climate models.

We also explore the relationship between $E_t/P$ and aridity index, unveiling a bell-shaped curve at the watershed scale, where the maximum $E_t/P$ ratio occurs at an aridity index between 2 and 3, corresponding to an $E_t/P$ ratio of around 0.5 to 0.58. These findings provide valuable insights into the intricate interplay between hydrological processes and environmental variables, shedding light on the complex dynamics of evapotranspiration in diverse watershed ecosystems.

## Appendix A

 **Table A1: Relative and absolute change in mean Et/E values due to changes in fast transpiration depth for the NARR dataset**

| Type | % Change in Et/E | | | Absolute change in Et/E | | |
|---|---|---|---|---|---|---|
| | 5 cm to 10 cm relative to 5 cm | 10 cm to 15 cm relative to 10 cm | 5 cm to 15 cm relative to 5 cm | 5 cm to 10 cm | 10 cm to 15 cm | 5 cm to 15 cm |
| Crops | 10.65 | 9.12 | 20.75 | 0.05 | 0.05 | 0.10 |
| Grass | 13.11 | 9.87 | 24.27 | 0.04 | 0.04 | 0.08 |
| Shrubs | 9.20 | 6.83 | 16.65 | 0.03 | 0.03 | 0.06 |
| ENF | 8.85 | 6.68 | 16.11 | 0.06 | 0.05 | 0.11 |
| DBF | 7.84 | 6.71 | 15.08 | 0.04 | 0.04 | 0.08 |
| MF | 4.91 | 4.33 | 9.45 | 0.03 | 0.03 | 0.05 |

**Table A2: Relative and absolute change in mean Et/E values due to changes in fast transpiration depth for the MODIS dataset**

| Type | % Change in Et/E | | | Absolute change in Et/E | | |
|---|---|---|---|---|---|---|
| | 5 cm to 10 cm relative to 5 cm | 10 cm to 15 cm relative to 10 cm | 5 cm to 15 cm relative to 5 cm | 5 cm to 10 cm | 10 cm to 15 cm | 5 cm to 15 cm |
| Crops | 10.56 | 8.11 | 19.52 | 0.06 | 0.05 | 0.11 |
| Grass | 12.95 | 8.51 | 22.56 | 0.04 | 0.03 | 0.08 |
| Shrubs | 8.03 | 5.97 | 14.48 | 0.03 | 0.02 | 0.06 |
| ENF | 9.47 | 6.03 | 16.08 | 0.07 | 0.05 | 0.11 |
| DBF | 8.76 | 7.51 | 16.93 | 0.05 | 0.05 | 0.10 |
| MF | 5.73 | 5.07 | 11.09 | 0.04 | 0.04 | 0.08 |

**Table A3: Relative and absolute change in mean Et/E values due to changes in fast transpiration depth for the GLEAM dataset**

| Type | % Change in Et/E | | | Absolute change in Et/E | | |
|---|---|---|---|---|---|---|
| | 5 cm to 10 cm | 10 cm to 15 cm | 5 cm to 15 cm | 5 cm to 10 cm | 10 cm to 15 cm | 5 cm to 15 cm |

| | relative to 5 cm | relative to 10 cm | relative to 5 cm | | | |
|---|---|---|---|---|---|---|
| Crops | 10.31 | 9.11 | 20.36 | 0.04 | 0.04 | 0.09 |
| Grass | 13.42 | 10.58 | 25.43 | 0.03 | 0.03 | 0.06 |
| Shrubs | 9.00 | 7.05 | 16.68 | 0.03 | 0.02 | 0.05 |
| ENF | 9.13 | 7.01 | 16.77 | 0.06 | 0.05 | 0.10 |
| DBF | 7.52 | 6.40 | 14.40 | 0.04 | 0.03 | 0.07 |
| MF | 4.99 | 4.40 | 9.62 | 0.03 | 0.03 | 0.06 |

**Table A4: Relative and absolute change in mean Et/E values due to changes in fast transpiration depth for the Zhang dataset**

| Type | % Change in Et/E | | | Absolute change in Et/E | | |
|---|---|---|---|---|---|---|
| | 5 cm to 10 cm relative to 5 cm | 10 cm to 15 cm relative to 10 cm | 5 cm to 15 cm relative to 5 cm | 5 cm to 10 cm | 10 cm to 15 cm | 5 cm to 15 cm |
| Crops | 10.20 | 8.48 | 19.54 | 0.05 | 0.04 | 0.09 |
| Grass | 13.73 | 10.39 | 25.54 | 0.04 | 0.04 | 0.08 |
| Shrubs | 8.29 | 6.35 | 15.16 | 0.03 | 0.02 | 0.05 |
| ENF | 10.50 | 7.70 | 19.01 | 0.07 | 0.05 | 0.12 |
| DBF | 8.80 | 6.98 | 16.40 | 0.06 | 0.05 | 0.10 |
| MF | 7.13 | 4.33 | 11.77 | 0.06 | 0.04 | 0.10 |

**Table A5: Relative and absolute change in mean Et/E values due to changes in fast transpiration depth for the SPLASH dataset**

| Type | % Change in Et/E | | | Absolute change in Et/E | | |
|---|---|---|---|---|---|---|
| | 5 cm to 10 cm relative to 5 cm | 10 cm to 15 cm relative to 10 cm | 5 cm to 15 cm relative to 5 cm | 5 cm to 10 cm | 10 cm to 15 cm | 5 cm to 15 cm |
| Crops | 9.78 | 8.60 | 19.22 | 0.04 | 0.04 | 0.07 |
| Grass | 14.23 | 10.40 | 26.11 | 0.04 | 0.03 | 0.07 |
| Shrubs | 9.56 | 7.53 | 17.82 | 0.03 | 0.02 | 0.05 |
| ENF | 9.99 | 7.57 | 18.32 | 0.06 | 0.05 | 0.11 |

| | | | | | |
|------|------|------|-------|------|------|------|
| DBF | 7.56 | 6.39 | 14.43 | 0.04 | 0.04 | 0.07 |
| MF | 5.29 | 4.68 | 10.23 | 0.04 | 0.03 | 0.07 |

**Table A6: Relative and absolute change in mean Et/E values due to changes in fast transpiration depth for the BESS dataset**

| Type | % Change in Et/E | | | Absolute change in Et/E | | |
|------|------|------|------|------|------|------|
| | 5 cm to 10 cm relative to 5 cm | 10 cm to 15 cm relative to 10 cm | 5 cm to 15 cm relative to 5 cm | 5 cm to 10 cm | 10 cm to 15 cm | 5 cm to 15 cm |
| Crops | 8.76 | 7.61 | 17.04 | 0.03 | 0.03 | 0.05 |
| Grass | 14.06 | 10.80 | 26.38 | 0.03 | 0.03 | 0.06 |
| Shrubs | 9.50 | 7.46 | 17.68 | 0.03 | 0.02 | 0.05 |
| ENF | 8.90 | 6.89 | 16.41 | 0.05 | 0.05 | 0.10 |
| DBF | 7.40 | 6.33 | 14.19 | 0.04 | 0.04 | 0.07 |
| MF | 4.87 | 4.29 | 9.37 | 0.03 | 0.03 | 0.06 |

**Acknowledgements**

This work was supported by Schmidt Sciences, LLC through the LEMONTREE (Land Ecosystem Models based On New Theory, obseRvations and ExperimEnts) project.

**Author contributions:** AH implemented the research ideas, designed and performed the experiments, analyzed the results, drafted the manuscript. XL conceived the research ideas, designed the experiments, analyzed the results, supervised the investigation, and wrote and finalized the manuscript. ICP initiated the research topic, analyzed the results, edited the manuscript.

**Competing interests:** The authors declare that they have no conflict of interest.

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
