# Peer review of "Insights into evapotranspiration partitioning based on hydrological observations using the generalized proportionality hypothesis"

_EGUsphere, 2025_

## Author Response (AR1)

We thank the Editor (Dr. Miriam Coenders-Gerrits) and the two reviewers (Reviewer 1 and Dr. Stephen Good) very much for their valuable comments and suggestions on our manuscript entitled "Insights into evapotranspiration partitioning based on hydrological observations using the generalized proportionality hypothesis". In the following, we provide a point-by-point response to each of the reviewers' comments. We revised our manuscript based on the reviewers' feedback and our responses accordingly. We are confident that the quality and clarity of our manuscript has been enhanced after the revisions.

**Editor (Dr. Miriam Coenders-Gerrits)**

After reading your manuscript, the two reviewer reports and your reply, I decided that major revisions are needed.

Firstly, the authors should better explain the rationale behind equation 3. Mostly this equation is coming from the GPH, but there are also additions by you. What are those? This is essential as this is the novelty you bring (as also commented on by reviewer #2).

**Response:** We have clarified the rationale behind equation 3 in the revised manuscript in lines 87-101.

Additionally, the definition of ET0 as lambda*ET needs some explanation (and references) as well.

**Response:** We have explained the definition and reasoning of $ET_0$ in the revised manuscript in lines 115-125.

I also think the manuscript would benefit if ET is properly defined. Instead of talking about ET and ET0, maybe better to use terms as Etot=Et+Ei+Es. Where Ei+Es equals ET0 (=non-stomatal evaporation) and Etot=ET.

**Response:** The terms have been redefined in the revised manuscript to be as follows:

$E$: evapotranspiration

$E_p$: potential evapotranspiration

$E_i$: interception

$E_t$: transpiration

$E_s$: soil evaporation

$E_0$: initial evapotranspiration

Furthermore, I agree with reviewer #2 that the constant lambda is confusing as it indeed is often used as the latent heat of vaporization. Better use any other character.

**Response:** To avoid confusion, we have replaced $\lambda$ with $k$.

Lastly, I am happy to read that the authors decided to remove the arbitrary and not-necessary separation between arid and humid.

Looking forward to your revised version.

**Reviewer 1**

This study proposes a method to estimate long-term average T/ET ratios based on the generalized proportionality hypothesis. The authors collect data from 648 watersheds across the US, which are divided into 6 dominant vegetation types. The results show that the T/ET patterns vary with different vegetation types, with shrubs and grasslands show lower ratios and forests show higher ratios. A bell-shaped curve between T/P and aridity index Ep/P is also observed. The study is on a topic of interest to the audience of HESS. I have the following comments that I hope the authors could address in their revision.

Specific comments:

1. The method described in section 2.1 needs to be more clear. Initial evapotranspiration $ET_0$ is described as $\lambda V_p$ in Ponce and Shetty (1995). $V_p$ is usually assumed to be the same as PET. But here, $ET_0$ is described as $\lambda ET$. This difference needs to be explained.

**Response:** Yes, it is true that Ponce & Shetty (1995a, 1995b) used $\lambda PET$ to represent $ET_0$, but others have also used $\lambda W$, where W represents the soil wetting (e.g., Tang & Wang, 2017; Wang & Tang, 2014) and $\lambda ET$ (e.g., Tang & Wang, 2017; Abeshu & Li, 2021) in the literature. In the end, $ET_0$ is the variable of interest, and it is adjusted by an estimated parameter, $\lambda$. Thus, whether it is represented by PET, W, or ET, it won't affect the GPH concept and equation, since $\lambda$ is a "scaling coefficient" which is estimated based on data. Different researchers have chosen to represent it in different ways. We chose $\lambda ET$ due to the interpretability of the $\lambda$ parameter in our case. This has been clarified in the revised manuscript lines 109-113.

In addition, equation 4 suggests that the amount of T is ET minus initial ET in humid regions. In other words, evaporation E is assumed to be the same as initial ET. $ET = E + T = ET_0 + T$. This assumption needs to be further discussed.

**Response:** In applying the Budyko curves or the GPH equations, assumptions are often made for some components of these equations in the literature. Our study is no exception. In particular, we have made our assumptions based on previous research as discussed below:

- Abeshu & Li (2021) considered $ET_0$ to include "evaporation from canopy interception and surface depression ponding and transpiration from shallow water storage (mostly in the unsaturated zone)". "Recall that $E_0$ corresponds to three primary sources where water is easily available for vaporization: Direct evaporation from interception (canopy and litter interception), direct evaporation from the soil surface and temporally stored water in surface depressions, and transpiration from the shallow root zone." "A more recent isotopic evapotranspiration partitioning

experiment on tallgrass prairie in the Great Plains of North America by Sun et al. (2021) found that the top 10cm soil layer is a major source of the total evapotranspiration during the initial drying periods. Like croplands, high water volume is extracted for transpiration from the topsoil layer; thus, the initial vaporization is a dominant component."

- Gerrits et al. (2009) considered "Interception is the evaporation from the entire wet surface, so not only the canopy, but also the understorey, the forest floor, and the top layer of the soil. Although the latter seems to have an overlap with soil evaporation, we distinguish them by the fact that soil evaporation refers to rainwater that is stored in the soil and is connected with the root zone (De Groen & Savenije, 2006). In this paper we assume that evaporation from the deeper soil is not significant or can be combined with evaporation from interception."

- Savenije (2004) stated: "Note that the wetted soil surface should not be considered part of the soil moisture that feeds the transpiration process. The wet surface (extending to several millimetres of soil depth) feeds back the intercepted water through direct evaporation and not via a delayed transpiration process. Even a stretch of dry sand, without vegetation, can intercept water. After a rainfall event a wet "crust" of soil is formed, underlain by dry sand, which dries out again within a day. This soil can intercept several millimetres of rainfall." "One can distinguish fast transpiration and delayed transpiration. Fast transpiration is from shallow rooted plants (typically grass and annual crops) with a time scale of less than a month; delayed transpiration is from deeply rooted plants (trees, shrubs, perennial crops), which have a time scale longer than a month. Fast transpiration only draws on the upper soil layer (until 50 cm depth), whereas delayed transpiration draws on deeper soil layers."

- Based on the above, we assume that $ET_0$ includes interception, bare soil evaporation, and a portion (f) of transpiration representing the fast transpiration (this assumption will now apply to both humid and arid regions). Thus, in both arid and humid regions, T/ET will be calculated as $(1-\lambda)/(1-f)$.

This has been clarified in the revised manuscript in lines 115-125.

In terms of arid regions, a parameter f is introduced. How is this f value determined?

**Response:** The following paragraph describing *f* was mistakenly deleted from the original manuscript and was added back in lines 132-146 with an update regarding how *f* is defined in arid and humid regions.

"Since $f$ represents the fast response of transpiration, we follow a similar approach to Abolafia-Rosenzweig et al. (2020) in defining the ratio of surface transpiration using root distribution in soil water stress. We additionally distinguish between energy- and water-limited regions by constraining energy-limited $f$ using the aridity index as displayed in equation (4):

$$f = r_{10} \times S \times f_{AI}$$

Where $r_{10}$ is the root percentage in the top 10 cm of the soil, $S$ is the soil moisture availability, and $f_{AI}$ represents impact of available energy. If the aridity index (AI) is less than 1, the region is energy limited. Thus, $f_{AI}$ = $AI$. If AI $\geq$ 1, then $f_{AI} = 1$. The rationale behind this is that when $AI < 1$, only a fraction of the transpiration from the top surface layer is quantified to be part of the fast components due to its energy limited nature.

The soil moisture availability represents the moisture availability in the root zone for root water uptake. Abolafia-Rosenzweig et al. (2020) calculated the soil moisture availability as a function of soil moisture, wilting point, and field capacity. To rely on hydrological observations instead of simulated or remotely sensed soil moisture, we assume the soil moisture availability to be the ratio between baseflow and total streamflow ($Q_b/Q$). This ratio can give an indication of water availability in the soil, and hence can be used to indicate soil moisture availability. Since we apply this method at the watershed scale, there may be multiple vegetation types in the same watershed, and therefore, we calculate a weighted value of $f$."

The Impact on the results due to the updated definition of *f*, described above, is limited. Figures that show larger differences compared to those in the manuscript are presented below. As illustrated, the differences are modest for Fig. 5 (now Fig. 7) and minor for Fig. 10 (now Fig. 12). These differences do not affect any of the previous conclusions.

[Figure]

Figure 7 (updated). T/ET values for the watersheds using data from the six datasets: NARR, MODIS, Zhang et al. (2010), GLEAM after rescaling, SPLASH, and BESS.

[Figure]

Fig. 12 T/P versus the aridity index for six datasets: (a) NARR, (b) MODIS, (c) Zhang et al. (2010), (d) GLEAM after rescaling, (e) SPLASH, (f) BESS.

2. Figure 3 shows the λ values according to different vegetation types. It would be helpful if the authors could also show a figure of *f* values according to different vegetation types.

**Response:** The following figure has been added in the revised manuscript (Figure 6) to show how *f* values vary among the different vegetation types.

[Figure]

3. Figure 7: It is surprising to see that LAI does not have a clear relationship with T/ET. Maybe the authors can divide the data points based on the vegetation types and see if there is a clear pattern.

**Response:** There seems to be a moderate linear relationship between LAI and T/ET when separated into arid and humid as shown in the updated figure below (added to the revised manuscript as Figure 9). Separate scatter plots for each vegetation type did not show significant relationships between LAI and T/ET.

[Figure]

4. In Table 6, T/ET values are similar across 6 different vegetation types, based on the mean values from the 5 selected reference methods. In Table 4, the T/ET values from the new method are more diverse, with lower values in grasslands and shrubs and higher values in forests. This difference between the new method and reference methods should be further discussed.

**Response:** The larger differences with the new method are primarily due to the large differences related to the PET data products used, while for the reference methods, they are all based on the ET flux data from the flux tower measurements. In addition, there are some other differences between the new method and reference methods, including different scales and different underlying theories. For example, the reference methods are at the flux tower scale, while the new method is at the catchment scale. We have highlighted these differences in the revised manuscript in lines 456-458.

5. Lines 404-405: The bell-shaped curve is not from the T/ET vs aridity index relationship. It is from the T/P vs aridity index relationship in Figure 10. The T/ET vs aridity index relationship shown in Figure 9a does not have a clear pattern.

**Response:** Yes, this is a typing mistake that has been adjusted in the revised manuscript (line 471).

**Reviewer 2: Dr. Stephen Good**

The submitted paper by Hassan and coauthors presents an approach to determine the transpiration (T) component of a watershed's long term hydrologic balance based on the idea of the Generalized Proportionality Hypothesis (GPH). They apply this approach to a large number of different catchments throughout the US. From this they determine some relationships between T as a fraction of ET or P and landcover, aridity, leaf area, soil moisture, as well as between T and the fraction of runoff derived from baseflow.

Overall, this subject is likely of interest to HESS readers, however there already exist a large number of published ET partitioning approaches and the case for this new method needs to be made very carefully. It is worth noting that the approach presented here is based on the long-term hydrologic balance of a basin, which differs from many prior studies, and importantly it evaluates the T/ET fraction in relation to runoff partitioning (i.e. the fraction of runoff derived from baseflow). Given that, I suggest the following refinements of this study are required prior to possible publication:

- Many ET partitioning studies have been conducted, so what new things have we learned about T/ET from this study? The content outlined in the abstract isn't particularly new or novel. T/ET's dependence on things like ecosystem type, aridity and LAI have been documented elsewhere before.

**Response:** The issue with existing ET partitioning study is that there are many different methods proposed in the literature, but with significantly varying ranges for T/ET values, even for similar sites using the same flux measurements. Our goal is to provide a new perspective into the partitioning of ET from a hydrological standpoint. Additionally, only Mianabadi et al. (2019) considered a hydrologically based T/ET. They estimated T/ET by modeling interception (which includes topsoil evaporation) as a daily threshold process (threshold is the interception storage capacity) and used rainfall distributions to upscale it to the monthly and then annual interception. Transpiration was modeled as a monthly threshold process based on net rainfall (precipitation minus interception), with the threshold being the soil moisture storage estimated based on a hydrological model, and upscaled it to annual transpiration via a rainfall distribution. T/ET is then calculated by assuming ET is interception plus transpiration, since topsoil evaporation is included in interception, and deeper soil and open water evaporations are neglected.

Our work provides a method of estimating watershed scale T/ET based on watershed balance and hydrological partitioning, a new method that differs from previous ones in the literature. Studies of T/ET at watershed scales also provide an important perspective that is useful for upscaling T/ET from field scale to watershed scale.

This has been clarified in the revised manuscript in lines 57-66.

-The authors need to much more clearly make the case for the validity of equation (3) and the variables its linked to. It is not obvious when reading this text why this equation should hold and/or how they relate to transpiration. The concepts introduced are not clearly described. The concepts of ET_0 and how it would or wouldn't compete with PET or baseflow are not defined. What does 'competition' mean in this sense. The assumptions inherent in (3) need to be spelled out and a conceptual schematic needs to be presented to the readers. While eq (3) is not clear, nor are the variables f or lambda clear. Why is 10cm chosen. In humid regions f is assumed to be zero, and f is nonzero elsewhere? But even in humid regions there is evaporative loss from the near surface. Also root distributions are not the same as evaporative losses from the near surface. Similarly, what the quantity lambda represents isn't presented clearly. Rearranging equation (4) gives: Lambda = 1 - T/ET*(1-f), so is this is effectively the non-stomatal evaporation fraction. And thus Lambda*ET = E?

**Response:** Our response to these comments will include two parts; 1) the validity of Eq. (3), and 2) the assumptions regarding $ET_0$. The clarifications below will be included in the text of the revised manuscript for better clarity.

**First: the validity of the equation:**

Eqs (1-3) (or the Generalized Proportionality Hypothesis (GPH)) have been previously established in the literature based on the observed relationships found by L'vovich (1979) and the later mathematical derivation (and generalization) by Ponce & Shetty (1995a, 1995b). The proportionality hypothesis of the SCS method was obtained based on observed data from a larger number of watersheds (USDA SCS, 1985), which was then generalized by Ponce and Shetty (1995) as Eq. (1) – that is what is called GPH equation. There are a large number of studies discussing Eqs. (1-3) from different perspectives in the literature.

The above cited references established a two-stage partitioning concept. That is to partition the annual precipitation over two stages: the first stage partitions precipitation into catchment wetting and surface runoff; and the second stage partitions wetting (W) into evapotranspiration (ET) and baseflow (Qb). Both stages of partitioning follow the generalized formula $\frac{X-X_0}{X_P-X_0} = \frac{Y}{Z-X_0}$, referred to as GPH, where $Z$ is the flux being partitioned into $X$ and $Y$, where $Z$ and $Y$ can increase unbounded, but $X$ has a maximum value $X_p$. There is a threshold behavior that occurs, where $Y$ is not observed until a portion of $X$ is fulfilled. This portion is called the initial abstraction $X_0$. The two-stage partitioning is well established, has been proved with thermodynamic principles (Wang et al., 2015), and has been extensively used in the literature in studies such as Sivapalan et al. (2011), Wang & Tang (2014), Chen & Wang (2015), Tang & Wang (2017), Abeshu & Li (2021). In our work, we use the second stage

partitioning to partition wetting into evapotranspiration and baseflow. This has been clarified in the revised manuscript in lines 87-101.

The schematic below has been added to the revised manuscript (Figure 1) for clarification.

[Figure]

**Second**: the assumptions regarding $ET_0$

L'vovich (1979) showed that for partitioning soil wetting into evapotranspiration and baseflow, there is a threshold value of wetting that occurs before any baseflow is generated, all of which is evapotranspiration. This threshold value (initial abstraction) is what we defined here as initial ET (ET0). Additionally, L'vovich (1979) observed that increased wetting results in ET approaching its maximum value PET, with baseflow increasing unbounded with increases in surface wetting. Ponce & Shetty (1995a, 1995b) derived the formulas describing this relationship. Abeshu & Li (2021) followed the same formulation but defining ET0 as $\lambda*ET$ instead of $\lambda*PET$ (defined by Sivapalan et al. (2011) and Ponce & Shetty (1995a, 1995b)). We follow the formulation of Abeshu & Li (2021) in their equation 2. In the literature, PET, W (wetting), or ET has been selected to represent $ET_0$ based on each study's variable of interest. In fact, whether PET, W (wetting), or ET is selected is not critical, since the selected variable, PET, W, or ET is adjusted by an estimated parameter, $\lambda$. Thus, $ET_0$ can be equal to $\lambda*PET$, $\lambda*PET$, or $\lambda*ET$, depending on the selected variable, but the $\lambda$ value would be different corresponding to the variable selected. In other words, whether one uses $\lambda*PET$, $\lambda*PET$, or $\lambda*ET$ to represent $ET_0$, it won't affect the GPH concept and equation, since $\lambda$ is a "scaling coefficient" which is estimated based on data.

In Abeshu & Li (2021), ET is defined as consisting of two components: initial ET ($ET_0$ as defined above) and continuing ET (ETc). We follow a similar definition as Abeshu & Li (2021). With similar assumptions regarding $ET_0$ and ETc, we can solve for transpiration. Specifically,

in Eq (3) we can solve for λ ($ET_0/ET$) using observations of ET, PET, W, and Qb. Then based on the assumptions regarding $ET_0$ and ETc, we can calculate transpiration.

Regarding the assumptions of $ET_0$, this depends on what is considered to be included in this threshold that occurs prior to baseflow generation. Our assumptions are based on the following quotes from the literature:

Abeshu & Li (2021) considered $ET_0$ to include "evaporation from canopy interception and surface depression ponding and transpiration from shallow water storage (mostly in the unsaturated zone)". "Recall that $ET_0$ corresponds to three primary sources where water is easily available for vaporization: Direct evaporation from interception (canopy and litter interception), direct evaporation from the soil surface and temporally stored water in surface depressions, and transpiration from the shallow root zone." "A more recent isotopic evapotranspiration partitioning experiment on tallgrass prairie in the Great Plains of North America by Sun et al. (2021) found that the top 10cm soil layer is a major source of the total evapotranspiration during the initial drying periods. Like croplands, high water volume is extracted for transpiration from the topsoil layer; thus, the initial vaporization is a dominant component."

Gerrits et al. (2009) considered "Interception is the evaporation from the entire wet surface, so not only the canopy, but also the understorey, the forest floor, and the top layer of the soil. Although the latter seems to have an overlap with soil evaporation, we distinguish them by the fact that soil evaporation refers to rainwater that is stored in the soil and is connected with the root zone (De Groen & Savenije, 2006). In this paper we assume that evaporation from the deeper soil is not significant or can be combined with evaporation from interception."

Savenije (2004) stated: "Note that the wetted soil surface should not be considered part of the soil moisture that feeds the transpiration process. The wet surface (extending to several millimetres of soil depth) feeds back the intercepted water through direct evaporation and not via a delayed transpiration process. Even a stretch of dry sand, without vegetation, can intercept water. After a rainfall event a wet "crust" of soil is formed, underlain by dry sand, which dries out again within a day. This soil can intercept several millimetres of rainfall." "One can distinguish fast transpiration and delayed transpiration. Fast transpiration is from shallow rooted plants (typically grass and annual crops) with a time scale of less than a month; delayed transpiration is from deeply rooted plants (trees, shrubs, perennial crops), which have a time scale longer than a month. Fast transpiration only draws on the upper soil layer (until 50 cm depth), whereas delayed transpiration draws on deeper soil layers."

Based on the above, we assume that $ET_0$ includes interception, bare soil evaporation, and a portion (f) of transpiration representing the fast transpiration (this assumption will now be applied to both humid and arid regions). Thus, in both arid and humid regions, T/ET will be calculated as (1-λ)/(1-f). Thank you for your suggestions on including *f* for the humid regions as well. This has been clarified in the revised manuscript in lines 115-125. Regarding how *f* is calculated, please see our description of it in our responses to your other comment after this one.

The Impact on the results due to the updated definition of *f*, described above, is limited. Figures that show larger differences compared to those in the manuscript are presented below. As illustrated, the differences are modest for Fig. 5 (now Fig. 7) and minor for Fig. 10 (now Fig. 12). These differences do not affect any of the previous conclusions.

[Figure]

Figure 7 (updated). T/ET values for the watersheds using data from the six datasets: NARR, MODIS, Zhang et al. (2010), GLEAM after rescaling, SPLASH, and BESS.

[Figure]

Fig. 12 T/P versus the aridity index for six datasets: (a) NARR, (b) MODIS, (c) Zhang et al. (2010), (d) GLEAM after rescaling, (e) SPLASH, (f) BESS.

-Baseflow: This method is highly dependent on baseflow separation. It is an interesting and different component of this analysis. However the basflow separation only includes one

single sentence. This method needs to be elaborated and explained more thoroughly. Furthermore the sensitivity of the method to this quantity needs to be clarified.

**Response:** We have clarified baseflow separation in the revised manuscript with the following text in lines 155-166:

"Streamflow data were retrieved from the US Geological Survey (USGS), and their corresponding baseflow magnitudes were estimated by separating it from the streamflow data using a one-parameter digital filter separation method (Lyne & Hollick, 1979). Filtering methods separate direct runoff and baseflow by differentiating between frequency spectrums of the hydrograph, where low frequency flow represents baseflow and high frequency represents the direct runoff which has rapid responses to precipitation. We used the widely used tool developed by Purdue University, Web-based Hydrological Analysis Tool (WHAT, Lim et al., 2010, 2005; https://engineering.purdue.edu/mapserve/WHAT, last accessed 25 Oct 2022), to separate baseflow from the observed streamflow. We set the value of the filter parameter to 0.925 which is within the suggested range. We did a sensitivity analysis (in a separate study) and used different filter values and methods available from WHAT, the results were similar. Since other methods such as Eckhardt (2005) require knowledge of hydrogeological conditions, we chose the one-parameter digital filter method due to its simplicity and constant parameter value, which produces plausible results (Eckhardt, 2008; Xie et al., 2020)."

SPECIFIC COMMENTS:

L15 – Its hard to follow why PET needs to be rescaled when you haven't introduced how PET is used your generalized proportionally approach.

**Response:** PET is shown in equations 2, 3, and 8. As stated in lines 82-84, the generalized proportionality hypothesis partitions a water quantity Z into its components X and Y, where X is bounded by its potential values Xp. In this case X is ET and Xp is PET. The problem with PET is that when you use different PET data products there are large differences in their values. Therefore, we introduce the rescaled PET, which takes advantage of the consistent ET/PET ratio from these data products and applies it to observed ET to obtain the rescaled PET. In doing so, we reduce the large variations involved in PET data products to some extent, while we keep each data set's internal characteristics represented by each product's own ET/PET ratio.

L22 –Good 2017 also confirmed this relationship at larger scales using remote sensing.

**Response:** Yes, that was an oversight. It has been added in the revised manuscript in line 22.

L84 – While I understand the terms ins equation (1) the underlying justification that the right hand side should be equal to the left hand side is missing. Why should these two terms be equal?

**Response:** As stated above in our response to the general comments, the generalized proportionality hypothesis has been extensively tested and used in the literature for the two-stage partitioning. We understand that this may not be clear enough in the manuscript, therefore have included the explanations provided to earlier comments above in the revised manuscript.

L88 – How would 'initial ET' compete with baseflow or PET? Please spell this out?

**Response:** This has been discussed in the earlier response to the general comments and has been made clearer in the revised manuscript.

L91- The use of lambda here may be confusing. Often times in evaporation studies lambda (or lambda ET) represents the latent heat of vaporization.

**Response:** We have used lambda since it is typically used to denote the portion of initial abstraction. Ponce & Shetty (1995a, 1995b), Sivapalan et al. (2011), Wang & Tang (2014), Chen & Wang (2015), Tang & Wang (2017), Abeshu & Li (2021) all used lambda to describe this portion. However, we have changed it to $k$ in the revised manuscript to avoid any confusion.

L92 – Why does this vary? What is the basis of this assumption? How is the breakpoint between Arid and Humid specified? Would it not be simpler to set 'f' as non-zero in humid zones and keep your method consistent across the aridity gradient? Note that even in humid regions there is some transpiration from the upper 10cm so f wouldn't be exactly 0.

**Response:** We agree with the reviewer that $ET_0$ should be treated consistently between arid and humid regions, since humid regions can also have some transpiration occurring from the topsoil. Therefore, the definition of T/ET will be the same across arid and humid regions and is defined as T/ET = (1-λ)/(1-f).

The definition of *f* was mistakenly deleted from the submitted manuscript, and has beem added back (lines 132 146) with a modification as follows:

"Since $f$ represents the fast response of transpiration, we follow a similar approach to Abolafia-Rosenzweig et al. (2020) in defining the ratio of surface transpiration using root distribution in soil water stress. We additionally distinguish between energy- and water-limited regions by constraining energy-limited $f$ using the aridity index as displayed in equation (4):

$$f = r_{10} \times S \times \boxed{f_{AI}}$$

Where $r_{10}$ is the root percentage in the top 10 cm of the soil, $S$ is the soil moisture availability, and $f_{AI}$ represents impact of available energy. If the aridity index (AI) is less than 1, the region is energy limited. Thus, $f_{AI}.= AI$. If AI $\geq$ 1, then $f_{AI} = 1$. The rationale behind this is that when $AI < 1$, only a fraction of the transpiration from the top surface layer is quantified to be part of the fast components due to its energy limited nature.

The soil moisture availability represents the moisture availability in the root zone for root water uptake. Abolafia-Rosenzweig et al. (2020) calculated the soil moisture availability as a function of soil moisture, wilting point, and field capacity. To rely on hydrological observations instead of simulated or remotely sensed soil moisture, we assume the soil moisture availability to be the ratio between baseflow and total streamflow ($Q_b/Q$). This ratio can give an indication of water availability in the soil, and hence can be used to indicate soil moisture availability. Since we apply this method at the watershed scale, there may be multiple vegetation types in the same watershed, and therefore, we calculate a weighted value of $f$."

L101 – How is f determined?

**Response:** Clarified in the response to the previous comment.

L103 – This is the first time 'simulation' has appeared, what is being simulated in this framework?

**Response:** We have clarified this in the revised manuscript in lines 129-130. The GPH equation is rearranged to be in terms of wetting as shown in equation (8). This equation is a function of hydrological observation and the lambda (now k) parameter. We also can calculate observed wetting from equation (7). Using observed and simulated wetting, we can find the optimal value of lambda (k) by maximizing the KGE of the simulated vs observed wetting as shown in equation (6).

L104 – The approach in eq (7) and (8) requires determination of Qb and Qd

**Response:** We stated in the manuscript that baseflow separation is performed to find Qb and Qd, and we will add clarification on that as per our response to the previous comment on baseflow separation.

L106 – Is it assumed that the fraction of roots in the upper 10 cm from the root distribution is equal to the 'f' value?

**Response:** The estimation of the f value has been clarified in the response to a previous comment.

L112 – This method is highly dependent on the baseflow separation. But only one sentence is given here. Please explain

**Response:** We have explained more on the baseflow separation method in the revised manuscript as per our response to your previous comment on baseflow.

L119 – Why is the 'soil moisture stress' calculated? Where is this used? Is there a justification for this definition of soil moisture stress?

**Response:** The soil moisture stress is calculated by $Q_b/Q$. This is clarified in the response to a previous comment.

L130 – What happened to the inset?

**Response:** Inset is at the bottom left corner of the figure, showing the watersheds in Alaska.

L137 - Which products from the ORNL DAAC were obtained?

**Response:** The dataset name is the Global Monthly Mean Leaf Area Index Climatology. This has been added to the revised manuscript in line 192.

L164 – Was the rescaling factor applied to the GLEAM ratio of ET/PET or just to the PET. This is unclear from how the authors have written this sentence. It seems the 0.7 is on the ET/PET but is only applied to the PET?

**Response:** Yes, the rescaling factor was applied to the ET/PET ratio, which is consequently used to calculate the rescaled PET. The GLEAM dataset has quite high ET/PET ratios, inconsistent with other datasets shown in Figure 2b (now 3b), so we rescaled GLEAM ET/PET by a factor of 0.7. Rescaled PET is then calculated as (PET$_{GLEAM}$/ET$_{GLEAM}$)*ET$_{watershed\ balance}$, where (PET$_{GLEAM}$/ET$_{GLEAM}$=0.7) is the reciprocal of the rescaled GLEAM ET/PET. We have clarified this in the revised manuscript in lines 220-228.

L170 – Make sure you clarify here (and elsewhere) that your observed values are from streamflow. Other observations, e.g. from remote sensing or flux towers are other possible interpretations if this isn't clarified.

**Response:** We agree that this may cause confusion, therefore, we have changed "observed ET" to be "watershed balance ET" in the revised manuscript to avoid confusion.

L267 – Does the linear trend presented here make sense. How can there be a non-zero T value when LAI = 0? Perhaps fitting a trend through the origin makes more sense? It looks fairly non-linear...

**Response:**. We agree that zero LAI should not produce non-zero T/ET, therefore we have redone Figure 7 (now Figure 9) such that the regression is fitted through the origin in the

revised manuscript. We additionally performed separate regressions for humid and arid regions, which presented moderate linear relationships between LAI and T/ET as shown below.

[Figure]

---

## Author Response (AR2)

We thank the Editor and the reviewer very much for their valuable comments and suggestions on our manuscript entitled "Insights into evapotranspiration partitioning based on hydrological observations using the generalized proportionality hypothesis". In the following, we provide a point-by-point response to the reviewer's comments. We have revised our manuscript based on the reviewer's feedback and our responses accordingly. We are confident that the quality and clarity of our manuscript have been enhanced after the revisions.

**Reviewer 1**

The authors did a good job on incorporating the reviewers' comments in the revision. The manuscript is improved. I still have a few comments that I hope the reviewers can address.

1. Lines 120-127: The introduction of parameter f and equation (4) is still not very clear to me. According to Lines 124-125, it is assumed that the remaining portion of E after deducting E0 is equivalent to the remaining portion of Et after deducting the portion f. This is a big assumption. This part needs better explanation from a physical point of view. The E after deducting E0 is usually called continuous evapotranspiration. Why is it equivalent to the amount of transpiration after deducting the fast transpiration from the top soil?

   Response: Based on the GPH definition, $E-E_0$ represents the evapotranspiration that competes with baseflow, whether we call it continuous evapotranspiration or not. Since shallow transpiration does not compete with baseflow, it should be excluded. According to the literature (as discussed in the manuscript lines 114-127 and in our previous responses), $E - E_0$ consists only of slow transpiration, because all other evaporative fluxes are included in $E_0$. From a physical standpoint, if we consider the evaporative fluxes that compete with baseflow, we would need to include slow transpiration and possibly deeper soil evaporation. However, the latter is typically neglected in GPH applications at long timescales, or combined with interception as in Savenije (2004), Gerrits et al. (2009), and Abeshu & Li (2021). Therefore, the only flux considered to compete with baseflow is the slow transpiration, leading to the relation:

   $$E-E_0 = E_{t\_slow}$$

   Equation 4 follows directly from this assumption. Since $E_0$ includes interception, evaporation from surface depression, topsoil evaporation, and shallow transpiration, the remainder of E must be the slow transpiration: $E-E_0=E_{t\_slow}$.

   For transpiration, we define fast transpiration as $E_{t\_fast}=f*E_t$, and thus slow transpiration as $E_{t\_slow}=(1-f)E_t$. Equating these two $E_{t\_slow}$ equations yields

   $$E-E_0 = (1-f)E_t$$

   By substituting $E_0$ with kE yields

   $$(1-k)E=(1-f)E_t$$

   which is equation 4.

2. Figures 4 and 5: The k values are pretty high. Some watersheds have k values higher than 0.9, which means over 90% of the evapotranspiration is considered as initial evapotranspiration. This is different from previous proportionality studies. For example, in Sivapalan et al. (2011) "Functional model of water balance variability at catchment scale: 1. Evidence of hydrologic similarity and space-time symmetry", the k value range is from 0 to 0.45. It seems to me that the physical meaning of initial evapotranspiration in this study is slightly different from the previous works. The authors should make clear statement about this difference. By the way, change "λ" to "k" in Figure 5.

Response: We acknowledge that the reported k values are relatively high. However, it is important to clarify what the initial E (i.e., $E_0$) represents in this context. Specifically, $E_0$ is the portion of evapotranspiration lost prior to the competition between E and baseflow. Many components of $E_0$ are not accessible to baseflow, and only moisture that reaches deeper soil layers (i.e., slow transpiration, as discussed in the previous point) participates in this competition. This explains why the resulting *k* values tend to be relatively high.

Regarding the Sivapalan et al. (2011) study, they adopted the formulation $E_0=kE_p$, which differs from other forms discussed in lines 110-112 of the manuscript. In contrast, we used $E_0=kE$. Since potential evapotranspiration (Ep) is typically much larger than actual evapotranspiration (E), it is reasonable that the k values reported by Sivapalan et al. are lower than those in our study. Furthermore, Abeshu and Li. (WRR, 2021) "Horton Index: Conceptual Framework for Exploring Multi-Scale Links Between Catchment Water Balance and Vegetation Dynamics" also reported similar or even higher k values (see their Fig 6), using the same formulation for $E_0$ as in our work ($E_0=kE$).

In Section 5, since parameter f is the new parameter introduced in this study, I recommend the authors to add in-depth discussion about the connection between f and other variables. Also, it would be helpful if the authors could do a sensitivity analysis on parameter f, to show how the values of f would affect the results of the proportionality equations.

Response: Since *f* represents the portion of fast transpiration, it affects how evapotranspiration is partitioned and may influence the $E_t$/E ratio. However, it does not affect the hydrological fluxes such as Q, Qb and Qd, because *f* is independent of k, and changes in k do not alter the values of *f*.

To address this point, we performed a sensitivity analysis on *f*, as suggested, and have incorporated the results into the manuscript (Section 4.4, Sensitivity of $E_t$/E to *f* values and Appendix A). We found only minor differences in the resulting Et/E when varying the fast response depth between 5 cm, 10 cm, and 15 cm. These differences fall within the range of uncertainty reported in the literature for evapotranspiration partitioning methods.